# Permafrost sensitivity to soil hydro-thermodynamics in historical and scenario simulations with the MPI-ESM

Félix García-Pereira<sup>1,2</sup>, Jesús Fidel González-Rouco<sup>1</sup>, Nagore Meabe-Yanguas<sup>1</sup>, Philipp de Vrese<sup>3</sup>, Norman Julius Steinert<sup>4</sup>, Johann Jungclaus<sup>3</sup>, and Stephan Lorenz<sup>3</sup>

**Correspondence:** Félix García-Pereira (felgar03@ucm.es)

#### Abstract.

The limited representation of soil processes in Land Surface Models (LSMs) contributes to the uncertainty in current state and future projections of permafrost change. In particular, insufficient LSM depths, coarse vertical discretizations, and the omission of hydro-thermodynamic coupling can strongly affect subsurface temperatures, active layer thickness (ALT), and permafrost extent. This work aims to gain knowledge on permafrost sensitivity to changes in the soil hydrology and thermodynamics in permafrost-affected regions. We explore the response of the Max Planck Institute Earth System Model (MPI-ESM) to historical and climate change scenario forcing using an ensemble of fully-coupled simulations under three configurations of permafrost hydrology: the standard model that will be taken as a *reference*, and two variants that generate rather *dry* or *wet* conditions across permafrost areas. Enhanced soil depth and vertical resolution within the LSM, JSBACH, were also incorporated globally to capture fine-scale thermodynamics and allow for deeper heat propagation.

Deepening the LSM reduces near-surface soil warming by about  $0.1 \,^{\circ}\text{C}$  dec<sup>-1</sup> in high radiative forcing scenarios, reducing permafrost retreat by up to 1.9– $3.1 \, 10^6 \, \text{km}^2$  by the end of the 21st century. However, the greatest influences are produced by the *dry* and *wet* configurations, which lead to distinct initial states, historical, and future evolution for permafrost temperatures (offset of  $3\,^{\circ}\text{C}$ ), active layer thickness (1–2 m) and permafrost extent ( $5\, 10^6 \, \text{km}^2$ ). These results indicate that inter-model spread in permafrost responses to climate change can be partly explained by differences in the representation of soil physics. Our findings underscore the importance of refining LSM hydrological and thermodynamic processes in ESMs, with implications for the assessment of risks related to carbon feedbacks and infrastructure vulnerabilities in Arctic regions.

#### 1 Introduction

Global mean near-surface air temperature has increased by more than 1 °C since the end of the 19th century due to an enhanced anthropogenic greenhouse effect (Chen et al., 2021; Gulev et al., 2021). This warming is nearly four times more intense in northern high-latitude areas due to Arctic amplification (Rantanen et al., 2022). As most of the arctic and subarctic land areas are underlain by permafrost (about 20.8 10<sup>6</sup> km<sup>2</sup>; Obu, 2021), i.e., perennially frozen soils, the intense and continuous surface

<sup>&</sup>lt;sup>1</sup>Geosciences Institute, IGEO (CSIC-UCM), Madrid, Spain

<sup>&</sup>lt;sup>2</sup>Faculty of Physical Sciences, Complutense University of Madrid (UCM), Madrid, Spain

<sup>&</sup>lt;sup>3</sup>Max Planck Institute for Meteorology, Hamburg, Germany

<sup>&</sup>lt;sup>4</sup>CICERO - Center for International Climate Research, Oslo, Norway

warming is propagated within the soil degrading the permafrost. As a result, an increasing amount of soil organic carbon that has accumulated in the region since the last glaciation (ca.  $1035 \pm 150$  PgC; Hugelius et al., 2014) can be decomposed by microbes in the soil (Hugelius et al., 2014; Turetsky et al., 2019). This, in turn, releases greenhouse gases into the atmosphere, amplifying the warming signal, i.e., the permafrost carbon feedback (Schuur et al., 2022). However, permafrost thaw does not only have implications at a global scale, but also has substantial regional impacts. Permafrost degradation causes soil compression and subsidence and the development of surface water bodies (Vonk et al., 2015), posing structural risks for infrastructures in the Arctic (Hjort et al., 2022). Moreover, thawing triggers contamination problems from mercury release to the atmosphere and ground waters (Schaefer et al., 2020; Langer et al., 2023). A good understanding of soil dynamics of permafrost areas is therefore crucial to foreseeing the global consequences of their degradation and evaluating the risks for Arctic communities and ecosystems (Parmesan et al., 2023).

The state of permafrost is monitored on relatively scarce and heterogeneously distributed sites across the Arctic. These sparse measures have been aggregated to create observational products that allow for analyzing the state of permafrost at a global scale. For instance, the Circumpolar Active Layer Monitoring Network (CALM; Brown et al., 2000) tracks the long-term response of the active layer thickness (ALT), i.e., the annual maximum permafrost thaw depth, while the Global Terrestrial Network for Permafrost (GTN-P; Biskaborn et al., 2015) also monitors permafrost temperatures at different levels. Despite their great value, these observational products offer a limited representation of permafrost areas due to their poor temporal and spatial resolution (Brown et al., 2002; Biskaborn et al., 2019; Obu et al., 2019), and the uncertainties associated with the thermal and hydrological properties of the terrain (Heuvelink, 2018).

State-of-the-art Earth System Models (ESMs) avert the spatial and temporal heterogeneity of the observational permafrost measurements. They allow for assessing the thermodynamic and hydrological responses of permafrost regions to the combined influence of natural and anthropogenic forcing (Andresen et al., 2020; Burke et al., 2020; Steinert et al., 2023). Near-surface temperatures and soil moisture resolved by Land Surface Models (LSMs) within ESMs can be subsequently used to derive estimates of permafrost extent (PE) and ALT evolutions (Burke et al., 2020; Steinert et al., 2023). ESM simulations of the industrial period stemming from different models of the Coupled Model Intercomparison Project Phases 5 (CMIP5; Taylor et al., 2012) and 6 (CMIP6; Eyring et al., 2016) agree on a severe decrease of PE and deepening of ALT in the industrial era (Koven et al., 2013; Burke et al., 2020; Steinert et al., 2023). However, there is considerable inter-model variability in the results, with differences in PE decrease of 10 million km<sup>2</sup> (Steinert et al., 2023) and mean ALT deepening of 2 m (Burke et al., 2020) by the end of the 21st century. These discrepancies are mainly due to different climate sensitivity of the ESMs and the different modeling of soil hydrology and thermodynamic processes by their LSMs, which are especially sensitive for Arctic regions (Slater and Lawrence, 2013). The representation of different processes related to surface and soil hydrology, such as snow insulation (Mudryk et al., 2020; Menard et al., 2021; Zhu et al., 2021), inclusion/omission of organic matter and moss insulation effects (Walvoord and Kurylyk, 2016), and vertical drainage impedance in the presence of ice (Andresen et al., 2020) can lead to relatively wetter or drier states in Arctic soils (de Vrese et al., 2023), with feedbacks for soil moisture availability and water cycle evolution under future warming Shared Socioeconomic Pathway (SSP) scenarios (Andresen et al.,

2020). Moreover, a more realistic representation of these processes within LSMs can lead to estimates of ALT and PE that are closer to the latest observation-constrained estimates (Chadburn et al., 2015; Peng et al., 2023).

On the other hand, the representation of subsurface thermodynamics in ESMs is influenced by the depth of the LSM. This LSM depth for the heat is defined by a zero-flux bottom boundary condition imposed at the deepest layer, which ensures energy preservation. Most ESMs include LSMs that are too shallow (Steinert et al., 2021a) to realistically accommodate long-term soil warming trends (González-Rouco et al., 2009, 2021) and heat storage (Cuesta-Valero et al., 2016; García-Pereira et al., 2024a; Steinert et al., 2024) under long-term anthropogenic warming. Including too shallow LSMs leads to overestimated soil temperature trends and variability, both globally (González-Rouco et al., 2021; Steinert et al., 2021b) and over permafrost regions (Alexeev et al., 2007; Hermoso de Mendoza et al., 2020). These temperature biases yield an overestimation of ALT deepening and PE loss with warming, being more pronounced for deep permafrost (Hermoso de Mendoza et al., 2020). This is especially sensitive for assessing the vulnerability of vast areas in Alaska and Siberia underlain by Yedoma, i.e., the deep organic-rich permafrost formed in the Pleistocene that contains around 400 PgC (Schuur et al., 2022), additional to the ca. 1100 PgC allocated in near-surface permafrost (Hugelius et al., 2014).

Previous studies using the JULES (Chadburn et al., 2015), the MATSIRO (Yokohata et al., 2020), the CLM4.5 (Hermoso de Mendoza et al., 2020), the JSBACH (Ekici et al., 2014; Steinert et al., 2021b), and the QUINCY (Lacroix et al., 2022) LSMs have explored the impacts of a better representation of thermodynamic and hydrological permafrost physics in standalone LSM simulations. Here we examine for the first time the potential role of Arctic soil wetness, LSM depth, and layer discretization in driving changes in PE loss and ALT deepening in the historical and SSP scenario periods using fully-coupled ESM simulations.

For that purpose, an ensemble of simulations with a modified version of the Max Planck Institute Earth System Model (MPI-ESM) was run. These modifications entail an increased LSM depth with a finer layer discretization and more realistic features in the permafrost soil hydrology of the MPI-ESM LSM component, JSBACH. In this paper, the sensitivity of permafrost dynamics to changes in soil hydrology is assessed under three configurations of the Arctic water-cycle. In order to assess how extreme these hydrologically divergent configurations are in representing PE, the results stemming from the ensemble are compared to CMIP6 estimates from Steinert et al. (2023). This comparison is extended to the historical PE using observation-constrained data coming from the European Space Agency Climate Change Initiative (Obu et al., 2021).

The article is structured as follows: an explanation on the physics of the modified version of JSBACH and MPI-ESM and the design and configuration of the experiments analyzed in this work is provided by Section 2, as well as of the metrics and data sets used for comparison; the response of the different configurations and vertical discretizations of JSBACH in terms of near-surface temperature, ALT, and PE are described and compared to observational estimates in Section 3; a discussion of the results and concluding remarks are presented in Section 4.

#### 2 Data and methods

### 2.1 Model description






For this work, an ensemble of fully-coupled simulations has been developed, using both the standard version of the MPI-ESM (MPI-ESM1.2-LR, Mauritsen et al., 2019) and a modified version that accounts for an improved representation of permafrost physics. MPI-ESM consists of the atmosphere model ECHAM (Stevens et al., 2013), and the ocean model MPIOM, including the ocean biogeochemistry module HAMOCC (Jungclaus et al., 2013). In the low-resolution LR setup, the ECHAM resolution is T63/L95 and MPIOM is GR1.5/L40, which corresponds to a 1.875° (1.5°) grid cell width over land (ocean). The LSM is JSBACH (version 3.2, Reick et al., 2021), which includes the simulation of the surface and subsurface thermodynamics and hydrology as well as vegetation dynamics (Giorgetta et al., 2013). The standard version of JSBACH (Fig. 1) does not include any coupling processes between soil thermodynamics and hydrology, which prevents soil ice from forming under freezing conditions. This limitation leads to overly warm soils and thawing depths in summer, contributing to a warm bias over high latitude continental areas. This issue was addressed by Ekici et al. (2014) and Steinert et al. (2021b) by incorporating the occurrence of water phase changes with temperature, and the dynamic calculation of soil thermal properties in previous versions of JSBACH. However, these changes were run only in standalone LSM simulations. A modified version of JSBACH allowing for hydro-thermodynamic soil coupling in permafrost regions in fully-coupled ESM simulations (JSBACH-HTCp hereafter, Fig. 2b) was introduced by de Vrese and Brovkin (2021) and de Vrese et al. (2023) and is used in this work. This version is only active across the areas covered by the mask shown in Fig. 2a (Hugelius et al., 2013, 2014), which represents the observed early 21st century permafrost extent. This allows for attributing the simulated changes in global climate exclusively to differences in the representation of processes in the northern permafrost domain.

A more realistic feature in JSBACH-HTCp in comparison with JSBACH is that both the soil water phase and thermal properties vary with temperature and moisture content, respectively. This allows for the existence of ice in Arctic soils, which is not represented by JSBACH. Other new hydrological features included with the JSBACH-HTCp comprise adapted interactions between soil hydrology and vegetation, with the inclusion of an organic topsoil layer, the possibility for supercooled water, an improved representation of percolation and drainage by including the effect of soil ice impedance, and the implementation of a simple wetland and a new multi-layer snow scheme. This multi-layer scheme explicitly resolves thermal conductivity within the snow cover. By contrast, JSBACH uses a single-layer scheme that does not resolve internal snow temperature gradients, but only affects the computation of surface albedo and the thermal properties of the top soil layers. A thorough description of all changes incorporated in JSBACH-HTCp can be found in de Vrese et al. (2023). Moreover, the JSBACH-HTCp allows for controlling some of the hydrological processes involved in the surface and near-surface soil water cycle within the permafrost areas (Fig. 2a). Two combinations of parameters (hereafter referred to as configurations) based on de Vrese et al. (2023) were used to generate two subsets of fully-coupled experiments with relatively drier or wetter states of the Arctic soil, DRY and WET, respectively. The DRY configuration (red labels and arrows in Fig. 2b) exhibits weak local moisture recycling due to low infiltration rates and drainage resistance. Consequently, the near-surface permafrost degradation results in progressively drier soils under this configuration. In contrast, the WET configuration (blue, Fig. 2b) assumes favorable infiltration coupled with

high drainage resistance, leading to moist soils. An intense local moisture recycling is also fostered by imposing a low resistance to evapotranspiration. Apart from the two subsets of three DRY and WET simulations with different vertical discretizations, a third subset of experiments was run using the physics of the original version of JSBACH (REF configuration hereafter) imposed globally as a benchmark to assess the impacts of implementing the WET and DRY configurations across the permafrost domain. Outside of the permafrost mask (Fig. 2a), the soil physics are the same as included in REF (Fig. 1) in all the experiments.

Furthermore, the JSBACH standard vertical scheme of five layers (5L) with a LSM depth of 9.83 m was deepened and refined near the surface. An enhanced vertical resolution accounts for a better representation of hydro-thermodynamic processes near the surface (Chadburn et al., 2015, see Table 1). Further, a more realistic deeper LSM depth permits an improved representation of subsurface temperature variability and land heat storage (González-Rouco et al., 2021; Steinert et al., 2021a; García-Pereira et al., 2024a). Thus, two new vertical discretizations are introduced: an 11-layer (11L) scheme with a LSM depth of 9.98 m, and a deepened 18-layer (18L) scheme with a LSM depth of 1391.48 m (Fig. 1). The 11L scheme substitutes the original 5L discretization by enhancing the vertical resolution while keeping an almost identical LSM depth. The 18L scheme shares the top 11 layers with the 11L scheme and adds seven additional layers to reach a depth of 1391.48 m. Imposing such a deep zeroflux bottom boundary condition guarantees that the bottom layer temperature is detached from surface temperature variations at multi-centennial and millennial time scales (García-Pereira et al., 2024a). A detailed description of the mid-layer  $(z_i)$  and LSM depths and thickness  $(dz_i)$  values of the 5L, 11L, and 18L schemes used in this work can be found in Table 1 and Fig. 1. The design of the layering follows an exponential function of the form  $dz_i = a_1 exp[a_2(i-a_2) + a_3] + a_4$  (Oleson et al., 2010; González-Rouco et al., 2021). In this work, the coefficients  $a_1$ ,  $a_2$ ,  $a_3$ , and  $a_4$  were set to 0.00198, 0.743, 83.77, and 0.0585, respectively. These values for  $a_{1-4}$  aimed at attaining virtually identical LSM depths for the 5L and 11L configurations, while maintaining a thickness of 0.065 m for the first soil layer in the three vertical schemes. The same  $a_{1-4}$  values were used to estimate the thickness of layers 12–18 for the 18L configuration (Table 1).

#### 2.2 Experimental setup: the MPIESM-PePE




Combining the three configurations of permafrost hydro-thermodynamics (WET, DRY, and REF) and the three vertical discretizations of the subsurface (5L, 11L, and 18L) yields a total number of nine experiments integrating the ensemble presented in this work, the MPI-ESM Permafrost Physics Ensemble (MPIESM-PePE hereafter). The study focuses on assessing the combined impact of the different configurations and vertical discretizations of the MPIESM-PePE on permafrost-related variables under industrial forcing conditions in 1850–2100. The experiment setup follows the scheme in Fig. 3. In all simulations the ocean component was initialized using a restart from a long-term ocean simulation stabilized to pre-industrial conditions. Each 5L and 11L experiment was run first in a piControl (PIC) phase of 50 years, which simulates a climate state compatible with the external forcing in 1850 and provides initial conditions for the historical runs (HIS; 1850 to 2014). These runs are subsequently continued into the 21st climate change stage under different radiative forcing scenarios (SSP1-1.9, SSP2-4.5, and SSP5-8.5; Riahi et al., 2017), following the standard forcings of CMIP6 (Eyring et al., 2016). However, subsurface temperatures at the deepest layers of the 18L simulations do not have enough time to converge to the near-surface temperature equilibrium state within a 50-year-long PIC (González-Rouco et al., 2021). Therefore, a pre-piControl (prePIC) phase of 100 years is run before

**Table 1.** Layer number (i), mid-layer depth ( $z_i$ ), LSM depth, and thickness ( $dz_i$ ) of each layer in the three MPIESM-PePE vertical discretizations used herein. Note that 5L corresponds to the vertical structure in the standard JSBACH version, and 11L and 18L correspond to the vertical resolution enhancement and extension introduced in this work (see also Fig. 1).

| i         | $z_i$ (m) | LSM depth (m) | $dz_i$ (m) |  |  |  |  |
|-----------|-----------|---------------|------------|--|--|--|--|
| 5L        |           |               |            |  |  |  |  |
| 1         | 0.03      | 0.06          | 0.06       |  |  |  |  |
| 2         | 0.19      | 0.32          | 0.26       |  |  |  |  |
| 3         | 0.78      | 1.32          | 1.00       |  |  |  |  |
| 4         | 2.68      | 4.13          | 2.81       |  |  |  |  |
| 5         | 6.98      | 9.83          | 5.70       |  |  |  |  |
| 11L / 18L |           |               |            |  |  |  |  |
| 1         | 0.03      | 0.06          | 0.06       |  |  |  |  |
| 2         | 0.18      | 0.30          | 0.24       |  |  |  |  |
| 3         | 0.42      | 0.54          | 0.24       |  |  |  |  |
| 4         | 0.66      | 0.78          | 0.24       |  |  |  |  |
| 5         | 0.92      | 1.06          | 0.28       |  |  |  |  |
| 6         | 1.21      | 1.36          | 0.30       |  |  |  |  |
| 7         | 1.59      | 1.82          | 0.46       |  |  |  |  |
| 8         | 2.14      | 2.46          | 0.64       |  |  |  |  |
| 9         | 3.03      | 3.60          | 1.14       |  |  |  |  |
| 10        | 4.67      | 5.74          | 2.14       |  |  |  |  |
| 11        | 7.86      | 9.98          | 4.24       |  |  |  |  |
|           |           |               |            |  |  |  |  |
| 12        | 14.33     | 18.68         | 8.70       |  |  |  |  |
| 13        | 27.66     | 36.64         | 17.96      |  |  |  |  |
| 14        | 55.42     | 74.20         | 37.56      |  |  |  |  |
| 15        | 113.49    | 152.78        | 78.58      |  |  |  |  |
| 16        | 235.28    | 317.78        | 165.00     |  |  |  |  |
| 17        | 490.91    | 664.04        | 346.26     |  |  |  |  |
| 18        | 1027.76   | 1391.48       | 727.44     |  |  |  |  |
|           |           |               |            |  |  |  |  |

PIC for the 18L experiments to equilibrate the subsurface vertical structure and depart from vertically uniform temperatures. To speed up temperature convergence, the mean subsurface temperature of the last 50 years of prePIC of layer 12 (ST12) is imposed as the initial condition for the deepest layers (ST13 to ST18) in the 18L PIC simulations. The rest of the STs are continued from the last timestep in prePIC.

#### 2.3 Permafrost evaluation metrics






#### 160 2.3.1 Permafrost temperature variability

Surface and subsurface permafrost temperature variability is characterized by the mean annual air temperature at 2 m (MAAT), mean annual ground surface temperature (MAGST), which is here considered as the near-surface temperature at a depth of 0.1 m, and mean annual subsurface temperature at a depth of 5 m (MAST 5 m). The MAGST value is taken at 0.1 m depth for two reasons: to state a common depth for comparison between the 5L, 11L and 18L simulations, which have different mid-layer depths from layer 2 downwards (Table 1); and to fully account for the change in soil properties when there is snow cover. The latter is due to the fact that REF considers that snow gradually replaces the top layers of the soil when calculating the heat capacity and thermal diffusivity (Reick et al., 2021), thus modifying near-surface heat transfer and temperature variability.

To address the temporal and spatial evolutions of MAAT, MAGST, and MAST 5 m, areal boxplots summarizing the distribution of these variables over the permafrost domain (permafrost domain boxplots herein, see permafrost mask in Fig. 2a) are used for all the members of the JSBACH-HTCp ensemble in different periods. Furthermore, to assess the insulation effects of the refined multiple-layer snow scheme and organic layer introduced by Ekici et al. (2014), the winter and summer temperature offsets are assessed, respectively. In this paper, the winter (summer) offset is computed as the ground surface temperature at 0.1 m depth (GST) vs. surface air temperature (SAT) differences (GST-SAT) in December-January-February (June-July-August), following a similar approach to Burke et al. (2020). Winter and summer offsets regulate seasonal ground—air coupling (Melo-Aguilar et al., 2018; de Vrese et al., 2023), so evaluating them helps identify model biases in the representation of present-day and future projections of permafrost temperature and ALT (Nitzbon et al., 2025).

### 2.3.2 Active layer thickness (ALT)

ALT is defined as the annual maximum thaw depth, which in northern high latitudes occurs in summer. This variable is computed by JSBACH as the annual maximum subsurface depth where the temperature exceeds the water freezing/melting point, i.e., 273.15 K. If the depth of the 273.15 K isotherm does not precisely coincide with a given mid-layer depth, JSBACH computes ALT by linearly interpolating between the temperatures of the two adjacent layers that are immediately warmer or cooler than this isotherm. For all vertical discretizations, the maximum physically resolvable ALT corresponds to the mid-layer depth of its bottom layer. To ensure comparability between schemes, ALT was capped to 7.86 m; any value above this threshold was set to 7.86 m. This limit matches the mid-layer depth of the bottom layer in the 11L configuration and thus represents the deepest meaningful ALT for that setup. To evaluate the temporal and spatial variability of ALT, the same type of boxplot analysis as described for MAAT, MAGST, and MAST 5 m in Section 2.3.1 was used.

Furthermore, the spatial progression of ALT deepening under industrial warming in the MPIESM-PePE is analyzed by evaluating the fraction of grid cells within the initial permafrost domain (Fig. 2a) that exhibit an ALT shallower than a given threshold depth. This approach yields vertical profiles with values ranging from 0 (no gridcells) to 1 (all gridcells comply with the condition ALT < depth). As a result of permafrost thawing and subsequent ALT deepening with warming, this profile is expected to displace leftwards, i.e., values become closer to 0; the opposite happens under permafrost formation with cooling

conditions. Finally, near-surface and deep permafrost degradation is evaluated in terms of ALT deepening assessing the time evolution of maps with gridcells having an ALT above 3 m and an ALT equal to the bottom mid-layer depth, respectively.

#### 2.3.3 Permafrost extent (PE)






Permafrost is defined as the ground that remains perennially frozen for two or more consecutive years (Obu, 2021). However, there is a wide variety of metrics to evaluate whether the soil is frozen or not and subsequently determine PE based on different criteria that consider soil thermodynamics, hydrology, or soil coupling to the atmosphere (Steinert et al., 2023). Airground temperature coupling and some of the thermodynamic-based definitions capture near-surface PE variations, whilst others represent long-term deep PE changes. Hence, as the diverse definitions of PE target different aspects of permafrost physics, they can yield considerably different estimates. The PE differences between the different definitions are even larger than the inter-model variability of the CMIP6 ensemble (Steinert et al., 2023).

Since the WET and DRY configurations represent extreme states of the hydrological cycle in the permafrost domain, encompassing a substantial portion of the uncertainty range within the ensemble of state-of-the-art Earth System Models (de Vrese et al., 2023), it is of interest to assess whether they also capture the range of variability in PE responses. To address this question, we try to minimize the uncertainty associated with the PE definition selection by estimating the PE evolution for the nine historical + SSP5-8.5 simulations of the MPIESM-PePE using two of these metrics; i) TTOP, which defines permafrost based on temperatures at the top of the permafrost layer being below 0 °C (Obu et al., 2019); and ii) ZAA, which considers permafrost existence when the temperature at the depth of zero annual amplitude is below freezing. TTOP has proven to be a comparatively better indicator of surface permafrost presence (Obu et al., 2019; Steinert et al., 2023), while ZAA is a more appropriate descriptor of deep permafrost occurrence than TTOP (Slater and Lawrence, 2013; Burke et al., 2020). The zero annual amplitude layer, defined as the subsurface layer where the annual temperature varies less than 0.1 °C, is generally located below 15 m, depending on the thermal diffusivity and the annual GST amplitude of the site (Alexeev et al., 2007; Cermak et al., 2014; Burke et al., 2020; García-Pereira et al., 2024b). Therefore, as the 5L and 11L subsurface temperature bottom layers lie above this depth (at 6.98 and 7.86 m, respectively; Table 1), the ground temperatures at the deepest layer for 5L, and 11L, and at layer 11 for the 18L configuration are selected. To achieve a seasonal temperature amplitude that is close to zero, the remaining annual cycle at these depths is filtered out by subtracting the annual cycle of monthly mean anomalies for 1850–1900. Subsequently, permafrost detection using ZAA is applied to the data.

In order to illustrate the magnitude of the differences between the variety of configurations and vertical discretizations used in this work, the results for the MPIESM-PePE are compared with the ZAA and TTOP PE estimates stemming from Steinert et al. (2023) for an ensemble of 34 CMIP6 HIS and SSP5-8.5 experiments. Also, data provided by the Permafrost Climate Research Data Package version 3 product of the European Space Agency Climate Change Initiative database (ESApCCIv3) are used as an illustrative reference. ESApCCIv3 delivers mid- to high-latitude Northern Hemisphere (north of 30°) gridded data of permafrost temperature and presence at yearly resolution for the period 1997–2018. The database is generated by using the CryoGrid permafrost model (Westermann et al., 2023) driven and constrained by satellite and reanalysis land surface temperature data. ESApCCIv3 is computed using ensemble modeling procedures to enhance accuracy and provide estimates

of uncertainty (see Obu et al., 2021). In this study, permafrost temperatures at multiple depths from ESApCCIv3 within the mask in Fig. 2a are used to estimate the temperature at the top of the permafrost layer, from which the TTOP PE is derived. For the ZAA PE, the annual mean temperature at a depth of 10 m is assumed to represent the zero annual amplitude layer and is used to compute the corresponding PE. It is important to remark that, while ESApCCIv3 provides a valuable pan-Arctic benchmark of permafrost state, it remains a observation-constrained model product with known biases in the representation of permafrost temperature, ALT, and PE (Obu et al., 2021). For this reason, this product is not used as ground truth of the current PE, but as a reference to illustrate the order of magnitude of PE differences between the different model versions used in this study. Therefore, the MPIESM-PePE vs. ESApCCIv3 PE comparison does not attempt to determine whether any of the MPIESM-PePE simulations are better or closer to the real Arctic permafrost state.

## 3 Results and discussion







#### 3.1 Temperature variability

The surface and subsurface temperature variability across the permafrost domain (Fig. 2a) in the MPIESM-PePE is shown in Fig. 4. The median values for MAAT, MAGST, and MAST 5 m (Fig. 4a) remain stable during the PIC period for all simulations except 5-layer WET (hereafter WET5) and 11-layer WET (WET11), which start from warmer initial conditions that gradually converge toward the colder state of the 18-layer simulation (WET18) by the end of the PIC. Note that the 5L and 11L simulations have a PIC duration of 50 years, whereas the 18L simulations are first equilibrated using a 100-year-long prePIC simulation and incorporating subsurface temperature corrections (Section 2.2). This strategy guarantees the departure from vertically uniform initial conditions for temperature at the beginning of the PIC and stable PIC temperatures for the deepest layers (not shown). For the near-surface temperatures, 18L simulations also show a stable PIC evolution and transition from the PIC to the HIS period. The thermal state established during the PIC period for all simulations remains largely unchanged until the second half of the 20th century, when global warming begins to intensify. Consequently, the early-historical (1850–1900) period serves as a reliable representation of the pre-industrial thermal conditions, and the late historical (1995–2014), mid-21st century (2041–2060), and late 21st century (2081–2100) conveniently represent the progressive temperature increase and subsequent warming standstill (SSP1-1.9), stabilization (SSP2-4.5), or acceleration (SSP5-8.5) depending on the SSP scenario (Fig. 4a).

Early-historical temperature differences between WET and DRY simulations are of about 3 °C for the permafrost domain median MAAT (Fig. 4b), MAGST (Fig. 4d), and MAST 5 m (Fig. 4f) values. This agrees with previous results by de Vrese et al. (2023), who demonstrated that WET permafrost soils exhibit higher latent heat flux values than DRY, enhancing evapotranspiration and subsequently increasing Arctic cloud cover. This, in turn, reduces the incoming solar radiation, resulting in colder temperatures for WET than DRY. Thus, the DRY configuration depicts the warmest state, with permafrost domain median (10th, 90th percentile) values of -6.0 (-11.2, 0.0) °C, -3.5 (-9.4, 2.4) °C, and -3.9 (-9.9, 2.3) °C for MAAT, MAGST, and MAST 5 m, respectively. The WET simulations show colder MAATs than REF, with a permafrost median (10th, 90th percentile) MAAT of -9.3 (-14.9, -2.4) °C for WET vs. -8.0 (-13.7, -0.7) °C for REF. However, these differences are reversed

at the ground surface and subsurface, with WET MAGST and MAST 5 m being on average 1.4 and 0.3 °C warmer than REF, respectively. This is due to the enhanced snow insulation effect in winter in the WET and DRY simulations, as a result of the multi-layer snow scheme that superseded the more simplistic single-bucket scheme included in REF (see Section 2.1). Temperature differences between the 5L, 11L, and 18L vertical schemes are at most around 0.4 °C. Therefore, there is a noticeable effect of changing the hydrological state with the DRY/WET configurations but no evidence of the vertical discretization affecting surface temperature variability, as reported in previous studies with the MPI-ESM by González-Rouco et al. (2021), and García-Pereira et al. (2024a).






Permafrost warming increases during the late historical period to reach median MAATs 1.3–2.2 °C higher than in the earlyhistorical (Fig. 4b), depending on the MPIESM-PePE member. Overall, the temperature increase for MAAT, MAGST, and MAST 5 m is more intense for the WET simulations, with median values being significantly higher in 1995–2014 than in 1850–1900 (1.5–2.2 °C, p < 0.05). This increment is slightly smaller for DRY, but also significant (1.3–1.9 °C). The combined effect of the 20th and early 21st temperature changes translates into decadal historical trends during 1850-2014 of about 0.1 °C dec<sup>-1</sup> for MAAT, MAGST, and MAST 5 m (Fig. 4c,e,g), which are greater for WET and REF than for DRY in every case. The historical warming is followed for all the MPIESM-PePE simulations by an overall permafrost median warming trend of around 0.8 °C dec<sup>-1</sup> in the 21st century for MAAT under SSP5-8.5, 0.4 °C dec<sup>-1</sup> under SSP2-4.5, and non-significant MAAT trends for SSP1-1.9 (Fig. 4c). Moreover, temperature variability increases with the intensity of the SSP scenario, with the interdecile range being around 0.1 °C dec<sup>-1</sup> for all simulations in the SSP1-1.9 scenario, and ranging from 0.3 to 0.4 °C dec<sup>-1</sup> for SSP5-8.5 simulations. Temperature trends are slightly smaller and spatial variability is enhanced at the ground surface. The increase in MAGST median trends and spatial variability varies with the configuration considered. For instance, while the MAGST trend values for the WET SSP5-8.5 simulations show a median (10th, 90th percentile) value of 0.70 °C dec<sup>-1</sup>  $(0.42-1.13 \text{ °C dec}^{-1})$ , the DRY simulations have a median of 0.76 °C dec<sup>-1</sup>  $(0.54-1.04 \text{ °C dec}^{-1})$ , and the REF median values increase to 0.83 °C dec<sup>-1</sup> (0.66–0.99 °C dec<sup>-1</sup>; Fig. 4e,f). The trends in the SSP5-8.5 experiments yield MAGST median values of permafrost by the end of the century that are the highest for the DRY configuration, with at least 90 % of the permafrost domain (Fig. 2a) experiencing MAGST values that are above the melting point in 2081–2100 (Fig. 4d).

Regarding the effects of having different vertical discretizations, differences between the 5L, 11L, and 18L schemes can be identified across several configurations and scenarios. However, the only signal that is robust and consistent across REF, WET, and DRY is the attenuation of the MAST 5 m warming trend under the higher radiative forcing scenarios (SSP2-4.5 and SSP5-8.5). Deepening JSBACH systematically reduces the SSP5-8.5 warming trend at 5 m depth by approximately 0.1 °C dec<sup>-1</sup> at 5 m depth, indicating that a deeper LSM dampens subsurface temperature responses to surface warming. This result is consistent with previous findings by González-Rouco et al. (2021) with the JSBACH in standalone experiments, who showed that increasing LSM depth results in weaker subsurface warming. However, the impact of changing LSM depth is not significant for MAAT or MAGST. Overall, there is a clear effect of changing the hydrological state in the DRY/WET configurations, but no evidence that vertical discretization affects surface temperature variability. This agrees with earlier conclusions from González-Rouco et al. (2021) and García-Pereira et al. (2024a).

A considerable reduction in permafrost temperature variability, especially noticeable for MAGST and MAST 5 m (Fig. 4d,f), starts at the beginning of the 21st century for the SSP5-8.5 scenario. Interdecile ranges are reduced by a factor of 1.08–1.21 for 2041–2060, and 1.14–1.49 for 2081–2100 with respect to 1995–2014, depending on the MPIESM-PePE configuration. This reduction is likely produced by a decrease in spatial near-surface temperature heterogeneity associated with the retreat of the snow cover area in winter. Fig. 5a explores this by showing the variability of the winter offset (see Section 2.3.1, boxplots) and snow depth (dots) in the permafrost domain for the same periods and scenarios as in Fig. 4. It can be seen that both snow cover and winter offset barely change in the historical, SSP1-1.9, and SSP2-4.5 periods, with very stable snow depth median values of 7.7 (REF), 7.2 (DRY), and 7.9 (WET) cm. A small decrease of around 1 cm is shown for SSP5-8.5 by the late 21st century. Even though snow depth values are quite similar between the three MPIESM-PePE configurations, winter offset shows remarkable differences between the REF, and the WET and DRY experiments. The permafrost domain median winter offset reaches 10–10.5 °C for WET and DRY, but is 8–9 °C lower in REF. This contrast extends to the spatial variability: the interdecile range (box width in Fig. 5a) ranges from 6–8 °C in WET and DRY, but is only of about 1 °C for REF. Moreover, WET winter offset values are systematically about 0.5 °C higher than DRY, which is consistent with WET being colder (Fig. 4) and having more snow cover. For the three configurations, thinning snow depth leads to less winter offset, being of 9.1 °C for WET and 8.4 °C for DRY by the end of the SSP5-8.5 period.






To better assess the dependency of the winter offset on snow depth for every simulation, a composite is computed for the December-January-February season by aggregating values of snow depth vs. winter offset for every grid cell and year during 1850–2100, including information of the historical period and the three SSP scenarios (Fig. 5b). Snow depth was binned at 0.2 cm intervals for depths below 2 cm and 1 cm intervals above, with the winter offset averaged within each bin. This approach allows for reducing the heterogeneity in snow depth values associated with its temporal and spatial variability. The WET and DRY simulations show a strong relationship between winter offset and snow depth, starting at 5 °C for a 2 cm snow cover and rising by 1 °C cm<sup>-1</sup> up to 10 cm of snow, followed by a slower rate of 0.5 °C cm<sup>-1</sup> for deeper snow cover values. In contrast, winter offset in REF remains significantly lower than in the WET and DRY simulations, with a maximum of only 1.8 °C at 7–8 cm of snow depth and an increase of just 0.25 °C cm<sup>-1</sup>. For snow depth values greater than 8 cm, the offset steadily declines at a rate of 0.07 °C cm<sup>-1</sup>. These differences are primarily due to the implementation of a multi-layer snow scheme in WET and DRY (Section 2.1), which enables more realistic thermal insulation. As the snow depth increases, new layers are added every 5 cm up to a maximum of five, beyond which the top layer continues to accumulate snow (Ekici et al., 2014). Each layer's temperature is explicitly computed, and energy transfer between layers occurs via conduction. As the thermal diffusivity of snow is low due to its low heat conductivity (in the range of  $0.25 \text{ WK}^{-1}\text{m}^{-1}$ ) and relatively high volumetric heat capacity (around 6 x 10<sup>5</sup> Jm<sup>-3</sup>K<sup>-1</sup>), it hinders heat transfer and effectively insulates the soil beneath the snow cover. REF lacks this vertical layering of snow cover. Instead, snow depth only modifies the surface albedo and thermal properties of the uppermost soil layers (Reick et al., 2021). As a result, REF winter offset values remain small and less responsive to snow depth, as opposed to WET and DRY. These contrasting behaviors and the importance of layered snow schemes for capturing snow insulation effects have also been highlighted in standalone JSBACH simulations (Ekici et al., 2014; Steinert et al., 2021b).

In addition to the relationship between snow depth and winter temperature offset in the WET and DRY simulations, Fig. 5b shows that a temperature offset also occurs in the absence of snow for these configurations. This offset, which is not present in REF, results from the insulation effect of the organic layer in the MPIESM-PePE simulations (see Section 2.1; de Vrese et al., 2023). The organic layer helps to smooth GST variability by dampening the response to SAT changes (Fig. 6). Specifically, it enhances the snow insulation effect in winter while mitigating GST increases in summer (Fig. 6a). As a result, the combined insulation effects of both the snow and the organic layer, along with latent heat exchanges from permafrost thawing and refreezing in summer and winter, lead to a reduced amplitude of the annual GST cycle in WET and DRY compared to REF (Fig. 6b). Thus, the permafrost domain median (10th, 90th percentiles) GST–SAT amplitude ratio in 1850–1900 is of 62 (50–72) % for WET11 and WET18, 65 (52–74) % for WET5, DRY11, and DRY18, 66 (53-75) %, but 95 (93–97) % for REF (Fig. 6c). The differences in the GST annual amplitude can influence ALT (Chadburn et al., 2015) and PE (Burke et al., 2020) variability.

## 3.2 Active layer thickness







The spatial and temporal changes for ALT in the MPIESM-PePE simulations are represented in Fig. 7. Both the ALT permafrost domain median, quartiles, and 10th and 90th percentiles are the smallest for WET during the historical and scenario periods. This is due to the permafrost temperatures being on average 3 °C and 1.5 °C colder than for the DRY and REF simulations (Fig. 4), respectively. However, despite REF having colder MAAT, MAGST, and MAST 5 m than DRY (Fig. 4b.d.f, respectively). REF median ALT is approximately 1 m deeper than DRY in the historical period. Figure 7b,c further shows ALT for REF is greater than for DRY for ca. 70 % of the gridcells in 1850–1900, and 60 % in 1995–2014. The higher values of ALT in the REF simulations can be explained by the combination of two factors. Although REF MAGST is lower than DRY in 1850–1900 (Fig. 4b), summer GST is considerably higher (Fig. 6b). This is due to the lack of organic layer insulation and the absence of water phase changes with temperature in this configuration. In the WET and DRY simulations, when soil temperatures rise above 0 °C further warming is partially prevented, since an amount of heat is absorbed by the ice melt as latent heat. This phenomenon is known as the zero-curtain effect (Outcalt et al., 1990; Mottaghy and Rath, 2006). Since the REF simulations do not include phase changes, summer warming is capable of penetrating deeper into the soil. The DRY configuration, in contrast, counterbalances the stronger temperature increase in summer with the zero-curtain effect, yielding lower ALT values than REF. Still, around 30 % of gridcells for DRY have a thicker ALT than for REF in 1850-1900, increasing to 40 % in 1995–2014 (note the value of the x-axis when DRY and REF curves intersect). This indicates the zero-curtain effect does not outweigh the greater heat propagation from higher MAGST below 4 (1850–1900, Fig. 7b) to 4.5 m (1995–2014, Fig. 7c), a depth at which the annual cycle is very attenuated (Fig. 6b). Moreover, permafrost soils are very shallow across large areas of Siberia in JSBACH (Steinert et al., 2021b), so the absence of deep soil ice in these areas, and therefore of a zero-curtain effect, promotes further ALT deepening as a response to the relative higher summer temperatures (Fig. 6b) in DRY than REF. The WET simulations, which yield colder permafrost temperatures (Fig. 4b,c,e) and more soil moisture (de Vrese et al., 2023), shows the highest fractions of gridcells holding permafrost for any value of ALT.

As permafrost warming intensifies in the 21st century under all scenarios, especially the SSP5-8.5, ALT depth increases for all simulations. That is evidenced from an extreme deepening of the ALT in 2081–2100 under the SSP5-8.5 scenario compared to 1995–2014 for all configurations (Fig. 7a, Table 2). These strong ALT variations are also illustrated by long-term changes in the vertical distribution of permafrost (Fig. 7d,e), including near-surface (i.e., shallower than 3 m, Fig. 8), and deep permafrost (Fig. 9). Even though most of the gridcells have a deeper ALT for REF than for DRY up to 1995–2014, the DRY simulations exhibit a greater deepening of ALT during the 21st century, with a proportion of gridcells that still hold permafrost in 2081–2100 for DRY around 20 % lower than for REF (Fig. 7e, Table 2).

**Table 2.** Permafrost domain median ALT values (in m) in 1850–1900, 1995–2014, and 2081–2100 in the SSP1-1.9, SSP2-4.5, and SSP5-8.5 scenarios for the nine simulations of the MPIESM-PePE. Fraction of gridcells that have permafrost in 1850–1900, 1995–2014, and 2081–2100 under the SSP5-8.5 scenario (see Fig. 7e) and its equivalence in deep PE in 2100 (in 10<sup>6</sup> km<sup>2</sup>, see Fig. 9) is also provided.

|       | Median ALT (m) |           |                            |           |          | Permafrost gridcells (%) |           |           | Deep PE (10 <sup>6</sup> km <sup>2</sup> ) |  |
|-------|----------------|-----------|----------------------------|-----------|----------|--------------------------|-----------|-----------|--------------------------------------------|--|
|       | HIS            |           | SSP1-1.9 SSP2-4.5 SSP5-8.5 |           | SSP5-8.5 | HIS                      |           | SSP5-8.5  | SSP5-8.5                                   |  |
|       | 1850-1900      | 1995–2014 |                            | 2081-2100 |          | 1850-1900                | 1995–2014 | 2081-2100 | 2100                                       |  |
| REF5  | 2.68           | 3.62      | 4.13                       | 5.79      | 6.98     | 96                       | 84        | 54        | 5.18                                       |  |
| REF11 | 3.01           | 3.73      | 4.35                       | 6.51      | 7.86     | 96                       | 85        | 57        | 4.46                                       |  |
| REF18 | 2.93           | 3.58      | 4.19                       | 5.83      | 7.72     | 96                       | 86        | 62        | 7.88                                       |  |
| WET5  | 0.78           | 0.85      | 1.05                       | 2.68      | 6.64     | 99                       | 93        | 64        | 7.69                                       |  |
| WET11 | 0.82           | 1.20      | 1.58                       | 3.78      | 7.80     | 100                      | 92        | 63        | 6.11                                       |  |
| WET18 | 0.81           | 1.23      | 1.33                       | 2.89      | 6.35     | 100                      | 92        | 70        | 9.09                                       |  |
| DRY5  | 2.57           | 2.68      | 3.83                       | 6.65      | 6.98     | 79                       | 73        | 36        | 1.01                                       |  |
| DRY11 | 2.52           | 3.67      | 4.94                       | 7.86      | 7.86     | 79                       | 72        | 36        | 1.45                                       |  |
| DRY18 | 2.17           | 3.26      | 4.46                       | 7.63      | 7.86     | 81                       | 74        | 44        | 2.93                                       |  |

The rapid ALT deepening for DRY is shown in Fig. 8. While pre-2000 deeper near-surface permafrost thaw in summer (purple to red) extends over a greater area for the REF simulations, DRY is dominated by a more intense near-surface and deep permafrost thaw during the 21st century (orange). In fact, Fig. 9 shows that in 2100, a much greater area still holds deep permafrost in REF than in DRY (see Table 2). The WET configuration yields shallower ALT and greater gridcell fraction values than REF and DRY for the whole duration of the simulation. The median WET ALT difference with respect to DRY increases during the 21st century (Fig. 7a), which agrees with higher MAGST (Fig. 4e) and MAST 5 m (Fig. 4g) trends, of 0.1–0.2 °C dec<sup>-1</sup>, for DRY than WET for the SSP5-8.5 scenario. The relatively colder temperatures results in the WET configuration still having permafrost in summer for around 70 % of the gridcells in 2081–2100. However, the SSP5-8.5 represents the most extreme warming scenario, and therefore the biggest response in ALT. The SSP1-1.9 permafrost ALT response is clearly weaker, with median values 0.2–1.2 m deeper in 2081–2100 than in 1995–2014 (Fig. 7a, Table 2). In turn, the SSP2-4.5 depicts a stronger ALT deepening occurrence than SSP1-1.9 in the 21st century, with an increase in ALT ranging from 2.0 (WET) to 4.2 (DRY) m (Table 2). In both SSP1-1.9 and SSP2-4.5 scenarios, the differences between the different MPIESM-PePE configurations are equivalent to those already mentioned for SSP5-8.5, but comparatively smaller.

Apart from the differences in ALT due to the MPIESM-PePE configuration, there are also notable differences produced by the LSM depth. All configurations show increasing ALT differences during the 21st century among the 5L, 11L, and 18L schemes. This is depicted by a greater increment in the ALT values for the shallow (5L, 11L) than the 18L simulations in 2041–2060, and 2081–2100 for the SSP5-8.5 scenario (Fig. 7a). The effect of the LSM depth is also evidenced by a growing offset in the gridcell fraction profiles during 2041–2060 (Fig. 7d), and 2081–2100 (Fig. 7e) between the shallow and 18L discretizations. Deepening the LSM also produces differences in deep permafrost that range from 1.4 to 3.4 10<sup>6</sup> km<sup>2</sup> in 2100, depending on the MPIESM-PePE configuration (Fig. 9, Table 2). The shallow vs. deep vertical scheme differences are greater when comparing 11L with 18L than between 5L and 18L, which indicates that using a coarse vertical resolution reduces the ALT bias produced by using a too shallow LSM depth (Chadburn et al., 2015). Nevertheless, it is increasing the LSM depth when resolving vertical heat diffusion the prevailing factor in damping long-term warming (González-Rouco et al., 2021; Steinert et al., 2021a; García-Pereira et al., 2024a), producing a relative cooling of 18L vs. 5L and 11L simulations.

Although LSM depth impacts the representation of deep permafrost, its near-surface effect is negligible. Fig 9b-e show no consistent differences in permafrost fraction of gridcells for historical ALT, and very small differences for gridcells having an ALT above 5 (2) m in 2041–2060 (2081–2100) under the SSP5-8.5 scenario. These small differences are also very hard to distinguish in terms of the evolution of near-surface permafrost existence, as shown in Fig. 8.

Regarding the spatial retreat pattern of permafrost in the MPIESM-PePE, both Figs. 8 and 9 show high agreement for the nine MPIESM-PePE simulations. All of them portray little to no presence of near-surface permafrost in Southern Siberia, Southern Canada, the Labrador Peninsula, and Scandinavia. The warming in the 20th and the first-half of the 21st century under the SSP5-8.5 scenario affects much more near-surface than deep permafrost, yielding its retreat in Southern Alaska, Western, and Southeastern Siberia. These regions lie near the southern margin of the permafrost zone, where comparatively warmer surface conditions make them more sensitive to permafrost retreat and active layer thickening under climate change (e.g., Biskaborn et al., 2019; Rantanen et al., 2022). By 2081–2100 for the SSP5-8.5 scenario, near-surface permafrost only remains in Northern Siberia for the WET simulations. In contrast, deep permafrost extent varies considerably with the MPIESM-PePE configuration and vertical discretization, being maximum for WET18, where permafrost in 2100 is still present in Northern Canada and a vast area across Central, Eastern, and Northern Siberia, and minimum for DRY5 and DRY11, where only a small patch of deep permafrost remains in the Central Siberian highlands.

#### 3.3 Permafrost extent estimates: multi-model comparison

The time evolution of PE for the nine simulations of the MPIESM-PePE in the three SSP scenarios are compared with the PE values derived from the ESApCCIv3 and CMIP6 PE estimates provided by Steinert et al. (2023) in Fig. 10. Both the MPIESM-PePE and the CMIP6 ensemble show a mostly constant permafrost area since 1850–1900 (see values in Table 3) until the last decades of the 20th century. WET and REF depict very similar early-historical PE values for both ZAA (Fig. 10a) and TTOP (Fig. 10b), in the range of 19.9–20.4 10<sup>6</sup> km<sup>2</sup> and 19.7–20.8 10<sup>6</sup> km<sup>2</sup>, respectively (Table 3). However, REF5 and REF11 ZAA, and REF11 TTOP estimates show more variability, with the periods of the largest PE values coinciding with the strongest volcanic cooling events in the historical period (e.g., an abrupt PE rise in the 1880s as a response to the Krakatoa

**Table 3.** Initial state of permafrost extent ( $PE_{1850-1900}$ , in  $10^6$  km<sup>2</sup>), and trends in the SSP1-1.9, SSP2-4.5, and SSP5-8.5 scenarios (in  $10^6$  km<sup>2</sup> dec<sup>-1</sup>) for the nine simulations of the MPIESM-PePE and the 10th, 25th, 50th, 75th, and 90th percentiles of the CMIP6 ensemble stemming from Steinert et al. (2023). Values for both ZAA and TTOP definitions. Non-significant (p > 0.05) SSP trends are marked in bold.

|                      | PE <sub>1850-1900</sub> |       | Trend PE <sub>SSP1-1.9</sub> |       | Trend PE <sub>SSP2-4.5</sub> |       | Trend PE <sub>SSP5-8.5</sub> |       |
|----------------------|-------------------------|-------|------------------------------|-------|------------------------------|-------|------------------------------|-------|
|                      | $(10^6~\text{km}^2)$    |       | $(10^6 \ km^2 dec^{-1})$     |       | $(10^6 \; km^2 dec^{-1})$    |       | $(10^6~{\rm km^2 dec^{-1}})$ |       |
|                      | ZAA                     | TTOP  | ZAA                          | TTOP  | ZAA                          | TTOP  | ZAA                          | TTOP  |
| WET5                 | 19.92                   | 20.33 | -0.11                        | -0.16 | -0.71                        | -0.68 | -1.49                        | -1.34 |
| WET11                | 20.29                   | 20.43 | -0.10                        | -0.08 | -0.75                        | -0.72 | -1.42                        | -1.39 |
| WET18                | 20.32                   | 20.37 | -0.09                        | -0.01 | -0.64                        | -0.63 | -1.15                        | -1.35 |
| DRY5                 | 15.22                   | 15.49 | -0.15                        | -0.10 | -0.84                        | -0.90 | -1.62                        | -1.65 |
| DRY11                | 15.12                   | 15.30 | -0.13                        | -0.10 | -0.73                        | -0.71 | -1.52                        | -1.57 |
| DRY18                | 15.87                   | 16.05 | -0.15                        | -0.11 | -0.67                        | -0.70 | -1.38                        | -1.58 |
| REF5                 | 20.52                   | 20.78 | -0.04                        | -0.06 | -0.61                        | -0.61 | -1.43                        | -1.40 |
| REF11                | 19.73                   | 20.01 | -0.08                        | -0.08 | -0.59                        | -0.57 | -1.45                        | -1.44 |
| REF18                | 20.03                   | 20.26 | -0.16                        | -0.15 | -0.55                        | -0.62 | -1.09                        | -1.36 |
| $CMIP6_{P10}$        | 6.68                    | 13.47 |                              |       |                              |       | -0.39                        | -1.21 |
| $CMIP6_{P25}$        | 12.91                   | 16.07 |                              |       |                              |       | -1.33                        | -1.60 |
| $\text{CMIP6}_{P50}$ | 17.20                   | 20.22 |                              |       |                              |       | -1.79                        | -1.57 |
| $CMIP6_{P75}$        | 21.01                   | 24.75 |                              |       |                              |       | -1.24                        | -1.49 |
| $CMIP6_{P90}$        | 27.66                   | 28.80 |                              |       |                              |       | -1.25                        | -1.36 |

eruption). In contrast, the DRY simulations depict an area around 5 10<sup>6</sup> km<sup>2</sup> smaller than the REF and WET ones for both ZAA and TTOP (Table 3) and less temporal variability.

In general, CMIP6 PE median values lie in between WET and DRY when the ZAA permafrost definition is used, and agree with WET and REF for the TTOP definition. As ZAA and TTOP definitions physically target deep and near-surface permafrost degradation, respectively, this result indicates that the CMIP6 yields larger near-surface than deep permafrost estimates, which are approximately 20 and 17 10<sup>6</sup> km<sup>2</sup>, respectively. The interquartile range in CMIP6 PE is similar for ZAA and TTOP, of around 8 10<sup>6</sup> km<sup>2</sup>, but the interdecile range is greater for the deep permafrost (ZAA), of around 20–22 10<sup>6</sup> km<sup>2</sup>. Considering the CMIP6 PE interdecile range as representative of the CMIP6 ensemble spread, the differences in the PE early-historical equilibrium state associated with imposing the WET or DRY conditions in northern high latitudes account for a quarter of it. This suggests that the representation of soil hydrology plays an important role in determining the current and future extent of large-scale permafrost. In fact, state-of-the-art ESMs have been claimed to have difficulties in reproducing nowadays PE, due to their limitations in representing land surface processes (Chadburn et al., 2015; Burke et al., 2020; Steinert et al., 2023), the non-inclusion of specific processes accelerating permafrost degradation at local scales, such as soil subsidence (Andresen et al., 2020), talik formation or thermokarst (Farquharson et al., 2019; Nitzbon et al., 2024), or the inability to account for

420

the permafrost-carbon warming feedback (Burke et al., 2013). These observation-simulation discrepancies in PE are suggested in Fig. 10, which exhibits that PE values from ESApCCIv3 in 1997–2019 are not only lower than the CMIP6 median, but also lower than all the MPIESM-PePE experiments, both for ZAA and TTOP. The differences are especially large for ZAA, which might be more sensitive to the aforementioned degradation processes enhancing deep permafrost losses that are not implemented in current LSMs, including JSBACH. However, the observation–simulation comparison presented here must be interpreted with caution. The ESApCCIv3 product is not a purely observational dataset but an assimilation product, and it has documented biases in the representation of permafrost temperature, ALT, and PE (Obu et al., 2021). Nonetheless, the comparison with ESApCCIv3 depicts the scale of changes in our different simulated states with respect to observation-constrained estimates. Also, the broad spread in PE values across ESApCCIv3, CMIP6 models, and MPIESM-PePE experiments illustrates the substantial uncertainty that still surrounds our knowledge of the current permafrost state, of which soil hydrology is only one contributing factor.

430

435

455

460

The steady state of PE in the 19th and most of the 20th century is followed by a decline that starts in the last quarter of the 20th century. The severity of the PE retreat mainly depends on the intensity of the SSP forcing (Fig. 10, Table 3). Hence, the low forcing SSP1-1.9 scenario shows a stabilization of PE after 2030 for the nine simulations of the MPIESM-PePE, with similar values for ZAA and TTOP DRY of around 12 106 km² and of 16 106 km² for WET and REF in their three vertical discretizations. SSP2-4.5 depicts a moderate loss until 2070 and a subsequent stabilization, with near-surface permafrost area values (TTOP, Fig. 10b) of 8 10<sup>6</sup> km<sup>2</sup> for DRY and 12 10<sup>6</sup> km<sup>2</sup> for WET and REF. For this scenario, there are moderate differences among 5L, 11L, and 18L simulations for ZAA, with PE values that differ by 1-2 106 km<sup>2</sup> by the end of the 21st century. Thus, DRY18 and WET18 PE are 8.6 and 13.1 10<sup>6</sup> km<sup>2</sup> in 2081–2100, but 7.3 and 11.1 10<sup>6</sup> km<sup>2</sup> for DRY11 and WET11, respectively. These differences in the deep permafrost are due to the increase in LSM depth better accommodating for the subsurface propagation of the 21st century warming (González-Rouco et al., 2021; Steinert et al., 2021a; García-Pereira et al., 2024a). Under the SSP5-8.5 scenario, the PE loss shows no signs of stabilising, either for the MPIESM-PePE or the CMIP6 simulations. The deep permafrost (Fig. 10a) is intensely degraded for the 5L and 11L simulations, with PE around 2.1, 6.4, and 6.9 106 km2 in 2081-2100, for DRY, WET, and REF, respectively. The degradation is less pronounced for the 18L experiments, with PE values of 4.0, 9.5, and 9.8 10<sup>6</sup> km<sup>2</sup>, respectively. These PE differences by the late 21st century are consistent with differences in trends, which are between 0.14 and 0.36 10<sup>6</sup> km<sup>2</sup> dec<sup>-1</sup> smaller for the 18L than for the 5L and 11L simulations (Table 3). In contrast, near-surface permafrost degradation, although also intense, does not show notable differences between the 5L, 11L, and 18L schemes, with DRY TTOP by 2081-2100 being very similar for the three discretizations (2.4–2.9 10<sup>6</sup> km<sup>2</sup>), WET and REF being in the range of 6.9–8.7 10<sup>6</sup> km<sup>2</sup>, and the CMIP6 median PE being slightly lower than WET and REF (6.1 10<sup>6</sup> km<sup>2</sup>). Both the 10th and 25th percentiles of the CMIP6 ensemble ZAA and TTOP PE are below than 1 10<sup>6</sup> km<sup>2</sup> for the last three decades of the 21st century, which entails that near-surface permafrost is almost fully thawed for at least a quarter of CMIP6 models by the end of the century for the highest radiative forcing scenario.

Figure 11 shows the spatial distribution of PE depicted in Fig. 10 for the historical and SSP5-8.5 scenario periods. At least 90 (75) % of CMIP6 models, all MPIESM-PePE experiments, and the observations display near-surface (deep) permafrost in Fig. 11b (Fig. 11f) across central, western, and eastern Siberia, southern Siberia highlands, northern Alaska and Canada,

and Nunavut mainland and island territories by the end of the historical period (1995–2014). There is a notable decline in PE since 1995–2014 for both definitions under the SSP5-8.5 scenario (Fig. 11c,d,g,h). Permafrost degradation is more intense in southern Siberia, Fennoscandia, and the Labrador Peninsula, warmer areas lying near the southern margin of the permafrost zone. By 2041–2060, only a quarter of the CMIP6 members and REF18, WET11, and WET18 still show some permafrost patches in southern Siberia (Fig. 11c,g). By the end of the century (2081–2100, Fig. 11d,h), the existence of permafrost is predominantly reduced to central and northern Siberia, the highlands in southern Siberia, northern Alaska, and the island territories of Nunavut. In these regions, at least half of CMIP6 models, and WET, REF, and DRY18 maintain some deep permafrost. In contrast, in DRY5 and DRY11 simulations near-surface and deep permafrost existence is reduced to Northern Siberia. In the WET18 and REF18 simulations, deep permafrost covers a much larger area in central Siberia and northern Canada than near-surface permafrost. Likewise, the DRY18 deep permafrost extends across vast areas in western Siberia, while near-surface permafrost is confined to eastern Siberia. Overall, the retreat pattern of PE for ZAA for 5L and 11L simulations is similar to TTOP, which indicates that shallow LSMs yield similar near-surface and deep permafrost degradation, whilst deep LSMs delay deep permafrost thawing as a response to intense surface warming (see also Fig. 10b).

## 3.4 Permafrost extent and LSM depth

To better disentangle the potential role of the LSM depth from soil hydrology in producing differences in near-surface and deep permafrost estimates, the ZAA and TTOP definitions are compared in Fig. 12. As already exhibited in Fig. 10, the WET and REF simulations depart from a higher PE value in 1850–1900 than the DRY for both PE definitions. CMIP6 TTOP PE values are very similar to WET and REF, and ZAA PE closer to DRY. In general, the CMIP6 median always shows larger TTOP than ZAA PE values (golden line in Fig. 12a being above the grey diagonal). In contrast, all the MPIESM-PePE 18L simulations show negative values of ZAA - TTOP. Even though the DRY18 simulation starts from a smaller PE than WET18 and REF18, and further diverges by 2100, the evolution for the all three is quite robust in yielding a larger ZAA PE. However, this is not the case for the 5L and 11L simulations, which show a very close agreement in the evolution of TTOP and ZAA permafrost for REF simulations, and slightly greater TTOP than ZAA PE values by the end of the 21st century for WET and DRY. This entails that the MPIESM-PePE simulations with a shallow LSM (see Table 1 or Figs. 1,2b) experience a more intense deep permafrost thawing, as previously noted in Figs. 10b and 11h. This factor also explains the asymmetry observed in the median and spread of TTOP and ZAA PE estimates from CMIP6 simulations (Figs. 10,12), as previously remarked by Steinert et al. (2023).

In fact, Burke et al. (2020) stated that having too shallow LSMs could result in an overestimation of deep permafrost thawing (i.e., an underestimation of PE with warming) using the ZAA metric. Since the zero-annual temperature amplitude layer is usually found below 15 m (e.g., García-Pereira et al., 2024b), LSMs imposing zero-flux bottom boundary conditions above this depth would not experience a full attenuation of the annual cycle at any level. For that reason, we apply a filter to remove the remaining annual temperature cycle in the lowest soil layer in the 5L and 11L simulations (see Section 2.3.3). However, this is not the only caveat about shallow LSMs. An insufficient LSM depth produces an overestimation of the temperature variability with depth, as shown in González-Rouco et al. (2021) with the same modeling framework. This results in the amplification of

the subsurface temperature trends associated with global warming. A LSM depth of at least 170 m has to be imposed to ensure the full thermal decoupling between the ground surface and the bottom layer response in simulations spanning the historical + SSP period (1850–2100). This is not the case for any LSM incorporated in CMIP6 ESMs, as represented by Fig. 12b, which depicts the differences in deep vs. near-surface PE. There is an apparent bias in ZAA - TTOP for CMIP6 models having a very shallow land component, i.e., with a LSM depth at 3–5 m (-3.1  $\pm$  3.7  $\pm$  10 for 1995–2014, and -6.7  $\pm$  3.6  $\pm$  10 km² for 2081–2100). This bias still exists for a subgroup of models having LSM depths of around 10–15 m, which includes MPIESM-PePE 5L and 11L simulations (-0.4  $\pm$  3.9  $\pm$  10 for 1995–2014, and -0.3  $\pm$  1.4  $\pm$  10 km² for 2081–2100). ESMs having relatively deep LSM components (> 40 m) exhibit negligible ZAA - TTOP differences and dispersion (-0.2  $\pm$  0.4  $\pm$  10 for 1995–2014, and 0.3  $\pm$  0.7  $\pm$  10 km² for 2081–2100), excluding IPSL-CM6A-LR (LSM depth of 90 m) in 2081–2100, which is an evident outlier (-7.3  $\pm$  10 km²). This deep cluster also includes WET18, DRY18, and REF18 simulations, which have deep vs. near-surface PE differences of 0.3, 0.2, and 0.0 for 1995–2014, and 1.5, 1.5, and 1.8  $\pm$  10 km² for 2081–2100, respectively. This reduction in ZAA-TTOP bias with increasing LSM depth is consistent with Steinert et al. (2023), who reported greater agreement in PE estimates by the end of the historical period across different permafrost definitions for CMIP6 models with deep LSMs. The little impact observed in ZAA - TTOP PE when including a deep LSM in a set of simulations with varying conditions of Arctic soil wetness confirms the relevance of LSM depth to yield unbiased PE estimates under global warming conditions.

## 4 Summary and conclusions

Permafrost degradation is a critical element for the sensitivity of the Earth system to climate change because of its significant impact on the terrestrial carbon cycle, society, and ecosystems. Despite observational data being important for monitoring the ongoing state of permafrost soils, their coverage is often spatially sparse and uneven in time at a global scale. ESMs can overcome these spatial and temporal limitations, as well as provide comprehensive insights into permafrost dynamics. However, state-of-the-art ESMs exhibit considerable inter-model variability in their projections of PE and ALT (Burke et al., 2020; Steinert et al., 2023). These differences are strongly influenced by their different representation of soil thermodynamics and hydrology (Andresen et al., 2020). This study assesses the impacts of changing both soil thermodynamics and hydrology on the Arctic permafrost dynamics for the first time in a fully-coupled ESM. This is achieved by modifying the LSM, including progressively refined and deeper vertical discretizations of the LSM (5L, 11L, and 18L) and generating comparatively wetter (WET) or dryer (DRY) conditions to quantitatively capture the permafrost conditions induced by changes in the LSM hydrological state (de Vrese et al., 2023).

Our results highlight the relevance of the representation of soil hydro-thermodynamic processes in determining permafrost temperature, active layer thickness, and permafrost extent. Differences in the permafrost water table can produce very different thermal states, with MAAT differences of more than 3 °C between WET and DRY configurations. However, differences in the GST seasonal variability are mainly produced by an improved representation of the snow and organic layer insulation effects, included in WET and DRY but not represented in REF. Higher summer temperatures, together with the omission of latent heat absorption due to permafrost thawing, leads to REF ALT being on average 1 to 2 m deeper in 1850–1900 than DRY and

WET, respectively. These differences decrease with warming, with the DRY simulation experiencing the greatest ALT values by the late 21st century under the SSP2-4.5 and SSP5-8.5 scenarios. Apart from the already mentioned factors, LSM depth plays also a substantial role in determining future ALT. Deepening the LSM, i.e., switching from the 5L or 11L to the 18L discretization, prevents the full degradation of the permafrost profile for 6–8 % of the gridcells within the permafrost mask, which is equivalent to an area of 1.5–2 10<sup>6</sup> km<sup>2</sup>.






The implications of soil hydrology and LSM depth are also remarkable for the evolution of the PE. Both TTOP, which physically represents near-surface PE, and ZAA, which tracks deep PE evolution, yield an initial PE of around 15  $10^6$  km<sup>2</sup> for DRY and 20  $10^6$  km<sup>2</sup> for REF and WET, respectively. These differences explain up to 76 % of the interquartile range for CMIP6 deep PE, 54 % for near-surface PE, and about a quarter of the interdecile range for both, highlighting the uncertainty in current PE estimates associated with the representation of soil hydrology. Considering that permafrost region contains  $1035 \pm 150$  PgC over  $17.6 \cdot 10^6$  km<sup>2</sup>, which corresponds to an areal mean of  $58.8 \pm 8.5 \cdot 10^{-6}$  PgC km<sup>-2</sup> (Hugelius et al., 2014), the differences between WET and DRY near-surface PE values would roughly amount for 250–340 PgC. This is around 40 % of the carbon currently stored in the atmosphere, a large fraction that indicates the importance of uncertainties in frozensoil hydrology for permafrost carbon storage, with high-impact implications for the permafrost–carbon feedback and, in turn, climate change.

Regarding the LSM depth, the 5L and 11L with respect to the 18L discretization show consistent PE differences in the three hydrological configurations, of 1.9–3.1 10<sup>6</sup> km<sup>2</sup>. This emphasizes the relevance of LSM depth in reducing deep permafrost degradation. In fact, CMIP6 models with LSMs deeper than 40 m show the smallest differences in ZAA – TTOP PE, indicating that ESM simulations with deep LSM vertical schemes are better at projecting the future degradation of deep permafrost and its sensitivity to surface warming. Furthermore, because ice-rich, carbon-bearing deposits (e.g., Yedoma) can extend well below 10 m, using deeper LSMs reduces bottom-boundary warming and leads to more realistic estimates of the vulnerability of these deep soil organic carbon reservoirs.

Finally, the comparison of PE values from the MPIESM-PePE with observation-constrained ESApCCIv3 estimates and other CMIP6 simulations illustrates how the range of permafrost response that can stem in one model from different assumptions in permafrost hydrology contributes to model uncertainties. Furthermore, observation-constrained products based on data assimilation, i.e., models constrained by observational evidence, are not free from model biases resulting from the decisions on parametrizations or physical processes included. The latter emphasizes the need fore more observations and assessment of uncertainties in observation-based products.

Data availability. Over-land MPIESM-PePE fields of MAAT, MAGST, MAST 5 m, ALT, and PE (for both the TTOP and ZAA definitions) are provided in the following Zenodo repository: https://doi.org/10.5281/zenodo.17454575 (last access: 27 October 2025). The permafrost spatial mask used in this work to perform the analyses is also available. Monthly-resolution outputs and other variables of interest can also be provided upon request. The CMIP6 PE data used in this study are publicly available and can be accessed at the following Zenodo repository:

https://doi.org/10.5281/zenodo.10103000 (last access: 22 April 2025). ESApCCI data are also publicly available and can be downloaded from https://catalogue.ceda.ac.uk/uuid/6e2091cb0c8b4106921b63cd5357c97c/ (last access: 22 April 2025).

Author contributions. FGP conceptualized the study. JFGR, FGP, and NJS prepared the MPIESM-PePE experiments, with the guidance of PdV. FGP processed the output data and performed the calculations. FGP and JFGR prepared the original draft of the manuscript. All the co-authors contributed to the analysis and discussion of the results and to the revision of the draft.

Competing interests. PdV is a member of the Editorial board of The Cryosphere.




Acknowledgements. This work has been developed within the frame of GreatModelS (project no. RTI2018-102305-B- C21) and SMILEME (project no. PID2021-126696OB-C21) from the Spanish Ministry of Science and Innovation (MICINN). We acknowledge the support from the Scientific Network PolarCSIC, funded by the Spanish Consejo Superior de Investigaciones Científicas (CSIC). We would also like to thank the Deutsches Klimarechenzentrum (DKRZ) for the resources granted by its ScientiDic Steering Committee (WLA) to run the MPIESM-PePE simulation ensemble under project ID bm1026. We also appreciate the efforts of the World Climate Research Programme's Working Group on Coupled Modeling, responsible for CMIP, and thank the contribution by Steinert et al. (2023) for deriving a valuable permafrost extent database for CMIP6 models. Additionally, thanks to Christian Hauck for editing this paper and Stefan Hagemann and two anonymous reviewers for their proof-reading and suggestions, which substantially contributed to improve the manuscript. Finally, we would like to have a special mention for our colleagues at the MPI-M Veronika Gayler, Tobias Stacke, and Helmuth Haak for their valuable technical support.

Financial support. This research work has been financially supported by SMILEME (project no. PID2021-126696OBC21), funded by call no. MCIN/AEI/10.13039/501100011033 from the Spanish Ministry of Science and Innovation (MICINN). FGP was funded by contract no. PRE2019-090694 of the MICINN and by the Ministry for the Ecological Transition and the Demographic Challenge (MITECO) and the European Commission NextGenerationEU (Regulation EU 2020/2094), through CSIC's Interdisciplinary Thematic Platform Clima (PTI-Clima). PdV was funded by the European Research Council under the European Union's Horizon 2020 research and innovation program as part of the Q-Arctic project (grant agreement No 951288).

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

**Figure 1.** Conceptual sketch illustrating the main hydrological and thermodynamic features of the JSBACH standard version (REF). The mid-layer depth values for the default 5-layer (5L) vertical discretization, with a LSM depth at 9.83 m, are given on the left part of the y-axis. The corresponding values for the two enhanced discretizations of 11 (11L) and 18 (18L) layers introduced by this work are also shown, with LSM depths at 9.98 and 1391.48 m, respectively. Note that bedrock is drawn deeper than typical values in permafrost regions to improve visual clarity.

**Figure 2.** HTCp changes applied to JSBACH standard version. (a) Northern Hemisphere (45–90°N) land permafrost affected areas (shaded light blue) in the late historical period (Hugelius et al., 2013, 2014), which are taken as the mask where the JSBACH-HTCp physics are implemented. Outside the area portrayed by the permafrost mask in (a), the REF (Fig. 1) configuration of JSBACH is used. (b) Conceptual sketch showing the features and the intensity (arrow size) of different thermodynamic and hydrological processes for JSBACH-HTCp DRY (red) and WET (blue) configurations. As in Fig. 1, bedrock is drawn deeper than typical values in permafrost regions to improve visual clarity.

**Figure 3.** Experimental setup of the MPIESM-PePE. Scheme of preindustrial control (PIC, black), historical (HIS, yellow), and 21st century climate change Shared Socioeconomic Pathway forcing scenario (SSP1-1.9, SSP2-4.5, and SSP5-8.5, red) simulations for each of the REF, WET, and DRY configurations of the MPIESM-PePE. This sequence was also run with different vertical discretizations of 5, 11, and 18 layers (5L, 11L, and 18L, respectively); see Table 1 and Fig. 1. PIC is conducted with conditions of 1850 for 50 years. The historical run is started at year 50 of PIC for the three vertical discretizations. To speed up subsurface temperature equilibrium in the vertical column for the 18L simulations, a prior preindustrial control (prePIC) phase of 100 years is run. The subsurface temperature mean for the last 50 years at the 12th layer for the prePIC is used to restart temperatures at layers 13 to 18 in the 18L configuration (see Table 1 and text for the details).

**Figure 4.** Temperature variability in the nine simulations of the MPIESM-PePE (see legend for colors) within the permafrost domain (Fig. 2a). (a) Time evolution (11-year moving average filtered outputs) of the median MAAT (dashed), MAGST (solid), and MAST 5 m (dotted line, °C) in the PIC, HIS, and different SSP scenarios (see Section 2.2). Spatial distribution of MAAT (b), MAGST (d), and MAST 5 m (f) given by permafrost domain boxplots of time averages for the selected periods highlighted in (a) with light yellow bands: early-historical (1850–1900), late historical (1995–2014), mid-21st century (2041–2060), and late 21st century (2081–2100) in the SSP5-8.5 scenario. The inner tick in the boxplot marks the median value within the permafrost mask area (Fig. 2a), the bottom and top lines the quartiles 1 and 3, and the bottom and top whiskers the 10th and 90th percentiles, respectively. MAAT (c), MAGST (e), and MAST 5 m (g) boxplots indicating the range of trends (°C dec<sup>-1</sup>) within the permafrost domain, for the historical (HIS, 1850–2014) and SSP1-1.9, SSP2-4.5, and SSP5-8.5 scenario simulations (2015-2100) are also portrayed.

**Figure 5.** Winter offset, defined as GST-SAT in DJF (see Section 2.3.1), and snow depth. (a) Winter offset (°C) permafrost domain boxplots for the nine simulations of the MPIESM-PePE (see legend for colors) in the different periods shown in Fig. 4b,c,e. The inner tick in the boxplot marks the median value within the permafrost domain (Fig. 2a), the bottom and top lines the quartiles 1 and 3, and the bottom and top whiskers the 10th and 90th percentiles, respectively. Snow depth median values (cm) for the same periods and simulations is shown with black dots. (b) Snow depth as a function of the winter offset within the permafrost domain for every simulation of the MPIESM-PePE.

**Figure 6.** Summer offset and annual temperature cycle amplitude. (a) Summer offset (°C) permafrost domain boxplots (see Section 2.3.1) for the nine simulations of the MPIESM-PePE (see legend for colors) in the different periods shown in Fig. 4b,d,f and Fig, 5. The inner tick in the boxplot marks the median value within the permafrost mask area (Fig. 2a), the bottom and top lines the quartiles 1 and 3, and the bottom and top whiskers the 10th and 90th percentiles, respectively. (b) Amplitude (°C) of the permafrost domain median GST (solid), SAT (dashed), and ST 5 m (dotted line) annual cycles during 1850–1900 for the different MPIESM-PePE simulations in (a). (c) Ratio of GST vs. SAT annual cycle amplitude (AA) during 1850–1900 given by permafrost domain boxplots as in (a).

**Figure 7.** (a) Active layer thickness (ALT, m) for the ensemble of MPIESM-PePE simulations in the different periods shown in Fig. 4b,d,f and 5a. Fraction of gridcells within the MPIESM-PePE permafrost mask with a summer ALT greater than the indicated depth in the y-axis for 1850–1900 (b), 1995–2014 (c), 2041–2060 (d), and 2081–2100 (e). For every plot, dark (light) gray dashed line indicates the mid-layer depth of the bottom layer for the 11L (5L) configuration. The mid-layer depth for the bottom layer of the 18L configuration lies outside the range shown in panels a–e and is therefore not included.

**Figure 8.** Near-surface permafrost presence with time for the combination of the three hydrological configurations (WET, DRY, REF) and vertical discretizations (5L, 11L, 18L). This is represented by the spatial distribution of the last simulation year in which ALT is smaller than 3 m for each MPIESM-PePE simulation. The time evolution for the historical and SSP5-8.5 experiments is shown in each case. Contour lines delimit areas where ALT does not reach 3 m within the temporal extent of the simulation. The colored distribution stands over the permafrost domain displayed in Fig. 2a.

**Figure 9.** Deep permafrost presence with time for the combination of the three hydrological configurations (WET, DRY, REF) and vertical discretizations (5L, 11L, 18L). This is represented by the spatial distribution of the last simulation year (HIS + SSP5-8.5) in which ALT is equal to the mid-layer depth of the bottom layer (ALT =  $z_n$ ) for each MPIESM-PePE simulation. For the 18L simulations, 11L JSBACH  $z_n$  was taken in order to allow for comparison. Contour lines delimit areas where ALT do not reach  $z_n$  within the temporal extent of the simulation. The colored distribution stands over the permafrost domain displayed in Fig. 2a.

**Figure 10.** Permafrost extent evolution according to ZAA and TTOP (see Section 2.3.3). (a) PE (10<sup>6</sup> km<sup>2</sup>) using the ZAA definition for the different MPIESM-PePE simulations (solid lines, see legend for colors), the CMIP6 ensemble (gold solid line for the median, and dark (light) golden shading for the interquartile (interdecile) range), and the observational product ESApCCIv3 (purple solid line). (b) Same as (a), but for TTOP PE metric. CMIP6 PE data is only available for the historical period and SSP5-8.5 scenario (Steinert et al., 2023).

**Figure 11.** Permafrost extent (PE) for different periods in the MPIESM-PePE and CMIP6 ensembles in the historical and SSP5-8.5 scenario, and observed historical extension. PE CMIP6 10th, 25th, 50th, 75th, and 90th percentiles spatial distribution (shading) for TTOP (a–d) and ZAA (e–h) for 1850–1900, 1995–2014, 2041–2060, and 2081–2100. The extent and distribution of PE for the different MPIESM-PePE experiments in the same periods are depicted with contour lines of different colors (see legend), whilst the observational ESApCCIv3 historical permafrost distribution in 1997–2019 is shown with a thicker purple contour line.

**Figure 12.** (a) Comparison of ZAA (x-axis) vs. TTOP (y-axis) PE estimates (10<sup>6</sup> km<sup>2</sup>) for the CMIP6 median and the different members of MPIESM-PePE ensemble (see Fig. 10). Solid lines depict the year-by-year evolution of ZAA vs. TTOP values. The symbols (see legend) illustrate the mean ZAA vs. TTOP values for different periods. ZAA and TTOP PE estimates are filtered with a 15-year moving average. (b) ZAA - TTOP PE differences (10<sup>6</sup> km<sup>2</sup>) in 1995–2014 (circles), and 2081–2100 (squares) for the 32 members of CMIP6 ensemble stemming from Steinert et al. (2023), the nine MPIESM-PePE simulations presented in this work, and the ESApCCIv3 observational PE data (only in 1997–2019). Data sources are sorted by their LSM depth (m, logarithmic x-axis). Note the break in x-axis between 100 and 1400 m.