# Peer review of "Permafrost sensitivity to soil hydro-thermodynamics in historical and scenario simulations with the MPI-ESM"

_EGUsphere, 2025_

## Referee Comment (RC1)

**Manuscript:** Permafrost sensitivity to soil hydro-thermodynamics in historical and scenario simulations with the MPI-ESM

**Major remarks**

Using the fully coupled MPI-ESM, the authors analysed the impact of different soil scheme setups on the simulation of various permafrost features and their trends under global warming conditions. The various setups of the JSBACH land surface scheme (LSM) comprise differences in soil hydrology, soil thermal dynamics and considered soil depths. They compared their results with results from the current CMIP6 ensemble and highlighted the importance of a deep LSM soil depth for more reliable projections of deep permafrost. Such a deep soil depth is currently not used in any of the existing operational ESMs.

The study is very interesting and provides some insights why CMIP6 models behave so differently in their simulation of the current and futures states of Arctic permafrost areas. The paper is generally well written, having an appropriate number of figures. However, there are a few points where the manuscript may be improved. This is especially the case for the abstract. Currently, the abstract starts with a paragraph comprising a general introduction to the permafrost under global warming topic, one sentence on exploring the response of MPI-ESM, a short method section and some general description of results. I believe that the abstract can be strongly improved by sharpening the text regarding research gap and objective of the paper as well as in summarizing the most important results. This can be done, e.g., by starting with about two sentences on the existing research gap followed by a precise statement of the research objective of the paper (e.g. gaining more knowledge on….). Here, exploring MPI-ESM response is a mean to fulfil this objective. Then, this can be followed by the method description and the results. For the results, I suggest pointing out the main qualitative results instead of listing numbers. I also would regard the relevant result that 'LSM depth plays a substantial role in determining future ALT.'

Partially, the manuscript text comprises a lot of number crunching (especially in lines 356-376). It should be checked whether this may be reduced by inserting appropriate tables.

In summary, I suggest accepting the manuscript for publication after minor revisions are conducted.

**Minor remarks**

In the following suggestions for editorial corrections are marked in *Italic*.

Line 89
… shown by *the latest* ESMs in *climate scenario* simulations.

Line 101-103
*For this work, an* ensemble of fully-coupled simulations *has been developed, using both* the standard … … physics.

Line 158-159
In *all simulations*, the ocean component was *initialized using a restart* from a long-term ocean simulation *stabilized* to pre-industrial conditions.

Please replace 'all the simulations' or similar instances by '*all simulations*' throughout the paper!

Line 175
… when there *is* snow cover.

Line 197
… 3 m and *an* ALT …

Line 208-210
To address this *question, we are* trying to minimize the uncertainty associated with the PE definition selection *by estimating* the PE evolution for the *nine* historical + SSP5-8.5 simulations of the *MPIESM-PePE using* two…

Fig. 4 caption – last sentence
… SSP5-8.5 *scenario* (2015-2100) *simulations.*

Line 316-318
*The thermal diffusivity of snow is low due to its low heat conductivity* (… …) *and relatively high volumetric heat capacity* (… …). *This hinders heat* transfer and effectively *insulates* the soil beneath the snow cover.

Line 324
… also *occurs in* the absence …

Line 326
… helps *to* smooth …

Line 343
*Here*, when soil …

Line 350
… Fig. 7c), *a* depth …

Line 352
… therefore of *a* zero-curtain …

Line 356-376
This part comprises the listing of too many numbers so that is not fluently readable. I suggest implementing a suitable table with those numbers and then refer to it in the text.

Line 364
… for DRY *is shown in* Fig. 8.

Line 375
… than *in* 1995-2014 …

Figure 8 and 9
The colour legend comprises too many colour steps so that it is partially difficult to visually separate those steps. See also comment to Fig. 11.

Figure 8 and 9 captions
… 18L). *This is represented by the spatial* distribution of the *final* simulation year *in* which ALT is … … for *each* MPIESM-PePE simulation.

Figure 8 caption
The time evolution for *the* historical and SSP5-8.5 experiments is shown in *each* case.

Line 388-389
*Nevertheless, it increases* the LSM depth when resolving vertical heat diffusion*, which is the prevailing factor* in …

Line 440-441
*Under the SSP5-8.5 scenario,* the PE loss shows no *signs of stabilising*, *either* for the MPIESM-PePE *simulations* or the CMIP6 *simulations*.

Figure 11 (and Figs 8 and 9)
Separating the lines is difficult in many panels, hence, the figure should be improved.

I recommend enabling a better comparability to Figs. 8 and 9. This, e.g., might be achieved by reducing the number of colour intervals, especially for the 21$^{st}$ century. For the latter, 20-years intervals are appropriate. Here, I would use clearly noticeable colour/interval borders that are consistent with the panels in Fig. 11, i.e. at 1900, at 2014, 2040, 2060, and 2080.

Figure 12 caption
Remove sentence: In this study …. 2.3.3). The caption should only describe the figure and its content.

Figure 12
I suggest plotting the MPI-ESM-PePE members on top so that they can be better distinguished. Identifying single symbols is not necessary for the CMIP6 models so that they may be partially covered.

Line 524
… profile *for 6*-8% of

Line 532-534
In fact, CMIP6 models *with* LSMs deeper than 40 m show the *smallest differences in ZAA-TTOP PE,* indicating that ESM simulations with deep LSM vertical schemes *are better at projecting* the future …

Line 551
… *Helmuth* Haak …

---

## Referee Comment (RC3)

In their manuscript, García-Pereira et al. perform historical and future simulations with the MPI-ESM, which was modified to account for a dry and wet configuration of Arctic soils, as well include various model soil depths. With the help of these simulations, the authors investigate the differences of these setups for soil surface temperature, active layer depths and permafrost extent for past and future scenarios. The work is an important effort and relevant for the journal Cryosphere. I believe the work is close to publication, but I have a few major comments that, in my opinion, should be addressed:

- I believe the manuscript can be somewhat stream-lined to stress novel results from this specific study. Some of the extensively discussed points are already addressed in other papers (e.g. aspects of the importance of freeze-thaw processes and soil hydrology for the Arctic heat and water balance). There is also some very technical information on model versions and reasoning for these changes since MPI-ESM of CMIP6 simulations (which is not applied here), which I believe is maybe not that interesting to readers not directly involved with the model (and would be anyways better fitted for a model development paper). I would probably just focus on what is present in the model versions presented here.

- Inclusion of observational constraints: The authors compare modelled permafrost extent to an observational-derived product (with moderate/poor agreement for both dry and wet configurations), but do not compare soil temperatures and active layer depths shown here with similar available products. I recommend consistently including ground temperature and active layer thickness observational products as well (https://climate.esa.int/en/projects/permafrost/), to better grasp the model performance in simulating soil temperature and active layer depths. Since these datasets are also based on thermal models and limitations should probably also be briefly mentioned.

- The simulation setups presented are idealized to potentially represent dry and wet soils, and it is unclear which set up is actually more realistic to represent the Arctic as a whole, both from an observational standpoint as well as in terms of process representation. Since there are both relatively dry and wet areas in the Arctic, would it be possible to combine the results from the wet and dry setup offline, using wetland coverage maps to weight every grid cell to give an estimated combined of permafrost thaw in the future?

- A bit of an open question: the authors stress the differences in permafrost extents by the different approaches/simulation setups, which I think is fair to grasp a magnitude of impact that can be communicated as a single number. However, the differences in permafrost extent likely are differences of areas of very deep permafrost (I assume mostly >10m depth?), which would be probably disconnected from surface fluxes and at which depths I would assume that mostly bedrock or mineral soil is found. Is the thawing in these depths really relevant for a discussion on climate impacts of

permafrost on surface hydrological feedbacks for the climate and carbon degradation? In my opinion the importance deepening of close-to-surface carbon-rich layers (such as the Yedoma domain) should be stressed, even if this is not reflected in a full permafrost retreat of the soil column.

- It is not completely clear to me from the abstract and conclusions, what the final message was regarding the effects of the difference model soil depth configurations on the trends of averaged active layer thickness and PE. Looking at the effects of various model soil depths in Figure 4, 6 and Figure 10, this effect seems small (and probably not worth it in terms of additional computational expenses for coupled models?).

- In terms of significance, would it be worth to include compute back-of-the-envelope calculations implications for carbon release from the permafrost (e.g. from typical spatial carbon contents?).

Minor Comments

L24 -> nearly four times ?

L48 I think PE is not defined anywhere.

L70 yedoma -> Yedoma

L74 Also the QUINCY model

Lacroix, F., Zaehle, S., Caldararu, S., Schaller, J., Stimmler, P., Holl, D., Kutzbach, L., & Göckede, M. (2022). Mismatch of N release from the permafrost and vegetative uptake opens pathways of increasing nitrous oxide emissions in the high Arctic. Global Change Biology, 28, 5973–5990. https://doi.org/10.1111/gcb.16345

L118 "An enhanced vertical resolution accounts for a better representation of hydro-thermodynamic processes near the surface (Chadburn et al., 2015)." -> Reference to your Table 1?

L188 "In case this depth does not specifically match a certain mid-layer depth value, JSBACH yields ALT using linear interpolation."

When would this be the case? I would imagine this would apply if a layer is partly frozen, but the authors explicitly define the >273.15 °C definition.

L293 Define winter offset in the text and why it is important.

L301 "Both results reflect weaker insulation in the standard snow model."
Is this because of different parameterizations of the snow model (heat constants?), or added processes?

Wouldn't it be possible to back this up with data from the simulation?

Figure 1. I know this is a conceptual figure, but the bedrock seems very deep? The bed rock in the Arctic is usually found around 1-2 m (unless in valleys).

---

## Author Comment (AC2)

Félix García-Pereira[1,2], Jesús Fidel González-Rouco[1], Nagore Meabe-Yanguas[1], Philipp de Vrese[3], Norman Julius Steinert[4], Johann Jungclaus[3], and Stephan Lorenz[3]

[1]Geosciences Institute, IGEO (CSIC-UCM), Madrid, Spain
[2]Faculty of Physical Sciences, Complutense University of Madrid (UCM), Madrid, Spain
[3]Max Planck Institute for Meteorology, Hamburg, Germany
[4]CICERO - Center for International Climate Research, Oslo, Norway

The authors would like to thank the reviewers for their constructive suggestions and the time they devoted to reading and evaluating the manuscript. We have tried to integrate all suggestions and think that the manuscript has improved with them. We do appreciate their contribution.

The next sections contain a detailed point-by-point response to the reviewers' comments. Comments are labeled by reviewers and in order of appearance, i.e. R2C3 is the third comment of reviewer 2. The original number by the reviewer is also preserved if it was given.

**1 Referee 1**

MAJOR REMARKS:

R1C1: REVIEWER'S COMMENT:

Using the fully coupled MPI-ESM, the authors analysed the impact of different soil scheme setups on the simulation of various permafrost features and their trends under global warming conditions. The various setups of the JSBACH land surface scheme (LSM) comprise differences in soil hydrology, soil thermal dynamics and considered soil depths. They compared their results with results from the current CMIP6 ensemble and highlighted the importance of a deep LSM soil depth for more reliable projections of deep permafrost. Such a deep soil depth is currently not used in any of the existing operational ESMs.

The study is very interesting and provides some insights why CMIP6 models behave so differently in their simulation of the current and futures states of Arctic permafrost areas. The paper is generally well written, having an appropriate number of figures.

AUTHORS' RESPONSE:

The authors welcome the positive perspective of the reviewer on the paper. We are grateful to Dr. Hagemann for the comments.

R1C2: REVIEWER'S COMMENT:

However, there are a few points where the manuscript may be improved. This is especially the case for the abstract. Currently, the abstract starts with a paragraph comprising a general introduction to the permafrost under global warming topic, one sentence on exploring the response of MPI-ESM, a short method section and some general description of results. I believe that the abstract can be strongly improved by sharpening the text regarding research gap and objective of the paper as well as in summarizing the most important results. This can be done, e.g., by starting with about two sentences on the existing research gap followed by a precise statement of the research objective of the paper (e.g. gaining more knowledge on. . . .). Here, exploring MPI-ESM response is a mean to fulfil this objective. Then, this can be followed by the method description and the results. For the results, I suggest pointing out the main qualitative results instead of listing numbers. I also would regard the relevant result that 'LSM depth plays a substantial role in determining future ALT.'

**AUTHORS' RESPONSE:**

**We strongly acknowledge the precise indications made by the reviewer to improve the abstract. Following the structure and rationale suggested by the reviewer, we included the following modified abstract to the revised version of the manuscript:**

The limited representation of soil processes in Land Surface Models (LSMs) contributes to the uncertainty in current state and future projections of permafrost change. In particular, insufficient LSM depths, coarse vertical discretizations, and the omission of hydro-thermodynamic coupling can strongly affect subsurface temperatures, active layer thickness (ALT), and permafrost extent. This work aims to gain knowledge on permafrost sensitivity to changes in the soil hydrology and thermodynamics in permafrost-affected regions. We explore the response of the Max Planck Institute Earth System Model (MPI-ESM) to historical and climate change scenario forcing using an ensemble of fully-coupled simulations under three configurations of permafrost hydrology: the standard model that will be taken as a *reference*, and two variants that generate rather *dry* or *wet* conditions across permafrost areas. Enhanced soil depth and vertical resolution within the LSM, JSBACH, were also incorporated globally to capture fine-scale thermodynamics and allow for deeper heat propagation.

Deepening the LSM reduces near-surface soil warming by about $0.1\ ^{\circ}\text{C dec}^{-1}$ in high radiative forcing scenarios, reducing permafrost retreat by up to $1.9\text{--}3.1\ 10^6\ \text{km}^2$ by the end of the 21st century. However, the greatest influences are produced by the *dry* and *wet* configurations, which lead to distinct initial states, historical, and future evolution for permafrost temperatures (offset of $3\ ^{\circ}\text{C}$), active layer thickness ($1\text{--}2$ m) and permafrost extent ($5\ 10^6\ \text{km}^2$). These results indicate that inter-model spread in permafrost responses to climate change can be partly due to differences in the representation of soil physics. This underscores the importance of refining LSM hydrological and thermodynamic processes in ESMs, with implications for the assessment of risks related to carbon feedbacks and infrastructure vulnerabilities in Arctic regions.

In the new abstract, however, we decided to keep the most relevant numbers regarding temperature, ALT, and permafrost extent differences, since we believe that they provide an order of magnitude of the changes that the reader might consider illustrative.

R1C3: REVIEWER'S COMMENT:

Partially, the manuscript text comprises a lot of number crunching (especially in lines 356–376). It should be checked whether this may be reduced by inserting appropriate tables.

AUTHORS' RESPONSE:

Following the reviewer's concern, we have included a new table gathering all numbers related to the characterization of the ALT (originally lines 356–376, Table 2, see lines 413–438 of the track changes document or lines 359–378 of the revised manuscript). Elsewhere, numbers have been eliminated where possible to ease the reading of the text (see also R1C17).

*MINOR REMARKS: In the following suggestions for editorial corrections are marked in Italic.*

R1C4: REVIEWER'S COMMENT:

'Line 89: ... shown by *the latest* ESMs in *climate scenario* simulations.'

AUTHORS' RESPONSE:

The text has been changed according to the reviewer's indication.

R1C5: REVIEWER'S COMMENT:

Line 101-103: *For this work,* an ensemble of fully-coupled simulations *has been developed, using both* the standard ... physics.

AUTHORS' RESPONSE:

The text has been changed according to the reviewer's indication.

R1C6: REVIEWER'S COMMENT:

Line 158-159: *In all simulations,* the ocean component was *initialized using a restart* from a long-term ocean simulation *stabilized* to pre-industrial conditions.

Please replace 'all the simulations' or similar instances by '*all simulations*' throughout the paper!

AUTHORS' RESPONSE:

The expression 'all the simulations' has been changed to 'all simulations' throughout the manuscript according

**to the reviewer's indication. The text in lines 158–159 has also been modified following the reviewer's comment (lines 194–196 of the track changes document).**

R1C7:  REVIEWER'S COMMENT:
  *Line 175: ... when there* is *snow cover.*

  **AUTHORS' RESPONSE:**
  **The text has been changed according to the reviewer's indication.**

R1C8:  REVIEWER'S COMMENT:
  *Line 197: ... 3 m and* an *ALT ...*

  **AUTHORS' RESPONSE:**
  **The text has been changed according to the reviewer's indication.**

R1C9:  REVIEWER'S COMMENT:
  **Line 208-210: To address this** *question, we are* **trying to minimize the uncertainty associated with the PE definition selection** *by estimating* **the PE evolution for the** *nine* **historical + SSP5-8.5 simulations of the** *MPIESM-PePE using* **two...**

  **AUTHORS' RESPONSE:**
  **The text has been changed according to the reviewer's indication.**

R1C10:  REVIEWER'S COMMENT:
  **...** *Fig. 4 caption – last sentence: ...* **SSP5-8.5** *scenario* **(2015-2100)** *simulations*.

  **AUTHORS' RESPONSE:**
  **The text has been changed according to the reviewer's indication.**

R1C11:  REVIEWER'S COMMENT:
  *Line 316-318: The thermal diffusivity of snow is low due to its low heat conductivity* (...) *and relatively high volumetric heat capacity* (...). *This hinders heat* **transfer and effectively** *insulates* **the soil beneath the snow cover.**

  **AUTHORS' RESPONSE:**
  **The sentence has been rephrased to:**

  "As the thermal diffusivity of snow is low due to its low heat conductivity (in the range of 0.25 $\text{WK}^{-1}\text{m}^{-1}$) and relatively high volumetric heat capacity (around 6 x $10^5$ $\text{Jm}^{-3}\text{K}^{-1}$), it hinders heat transfer and effectively insulates the soil beneath the snow cover."

following the reviewer's indication while keeping the original sense of the sentence (lines 373–375 of the track changes document).

R1C12: REVIEWER'S COMMENT:

Line 324: ... also *occurs in* the absence...

AUTHORS' RESPONSE:

The text has been changed according to the reviewer's indication.

R1C13: REVIEWER'S COMMENT:

Line 326: helps *to* smooth...

AUTHORS' RESPONSE:

The text has been changed according to the reviewer's indication.

R1C14: REVIEWER'S COMMENT:

Line 343: *Here*, when soil ...

AUTHORS' RESPONSE:

Following the indication of the reviewer, we modified the text as follows:

"In the WET and DRY simulations, when soil temperatures rise above 0 °C further warming is partially prevented, since an amount of heat is absorbed by the ice melt as latent heat".

R1C15: REVIEWER'S COMMENT:

Line 350: ... Fig. 7c), *a* depth ...

AUTHORS' RESPONSE:

The text has been changed according to the reviewer's indication.

R1C16: REVIEWER'S COMMENT:

Line 352: ... therefore of *a* zero-curtain ...

AUTHORS' RESPONSE:

The text has been changed according to the reviewer's indication.

R1C17: REVIEWER'S COMMENT:

Line 356-376: This part comprises the listing of too many numbers so that is not fluently readable. I suggest implementing a suitable table with those numbers and then refer to it in the text.

**AUTHORS' RESPONSE:**

**The large amount of numbers included in lines 356–376 concerning the evolution of ALT has been eliminated from the main text, and gathered into a new table (see Table 2 in the track changes or in the revised document), following the reviewer's indication (see also R1C3).**

R1C18: REVIEWER'S COMMENT:

Line 364: ... for DRY *is shown in* Fig. 8.

**AUTHORS' RESPONSE:**

**The text has been changed according to the reviewer's indication.**

R1C19: REVIEWER'S COMMENT:

Line 375: ... than *in* 1995–2014 ...

**AUTHORS' RESPONSE:**

**The text has been changed according to the reviewer's indication.**

R1C20: REVIEWER'S COMMENT:

Figure 8 and 9: The colour legend comprises too many colour steps so that it is partially difficult to visually separate those steps. See also comment to Fig. 11.

**AUTHORS' RESPONSE:**

**Following the reviewer's suggestion, we have reduced the number of colors in Figs. 8 and 9 from 16 to 10 to enhance visibility. To enable a more clear comparability with Fig. 11 (see R1C25), we have reduced the number of color bins in 2000–2100 from 10 to 5. This results in 1850–1900, 2041–2060, 2081–2100 periods exactly coinciding with specific color bins, and 1995–2014 being approximately represented by the color bin of 2000–2020.**

R1C21: REVIEWER'S COMMENT:

Figure 8 and 9 captions: ... 18L). *This is represented by the spatial* distribution of the *final* simulation year *in* which ALT is ... for *each* MPIESM-PePE simulation.

**AUTHORS' RESPONSE:**

**Both captions have been changed according to the reviewer's indication.**

R1C22: REVIEWER'S COMMENT:

Figure 8 caption: The time evolution for *the* historical and SSP5-8.5 experiments is shown in *each* case.

**AUTHORS' RESPONSE:**

**The text has been changed according to the reviewer's indication.**

R1C23: REVIEWER'S COMMENT:

Line 388-389: *Nevertheless, it increases* the LSM depth when resolving vertical heat diffusion, *which is the prevailing factor* in ...

**AUTHORS' RESPONSE:**

**Following the indication of the reviewer, we modified the text as follows (lines 447–448 of the track changes document):**

"Nevertheless, it is increasing the LSM depth when resolving vertical heat diffusion the prevailing factor in damping long-term warming ..."

R1C24: REVIEWER'S COMMENT:

Line 440-441: *Under the SSP5-8.5 scenario*, the PE loss shows no *signs of stabilising, either* for the MPIESM-PePE *simulations* or the CMIP6 *simulations*.

**AUTHORS' RESPONSE:**

**The sentence has been changed according to the reviewer's indications.**

R1C25: REVIEWER'S COMMENT:

Figure 11 (and Figs 8 and 9): Separating the lines is difficult in many panels, hence, the figure should be improved.

I recommend enabling a better comparability to Figs. 8 and 9. This, e.g., might be achieved by reducing the number of colour intervals, especially for the 21st century. For the latter, 20-years intervals are appropriate. Here, I would use clearly noticeable colour/interval borders that are consistent with the panels in Fig. 11, i.e. at 1900, at 2014, 2040, 2060, and 2080.

**AUTHORS' RESPONSE:**

**We agree with the reviewer's concern about the distinguishability of the lines in the different panels of Fig. 11. We believe that the lines are difficult to distinguish because the individual maps are too small. To increase the size of the maps while keeping the overall figure size unchanged, we have rearranged the display of the maps, sorting them into two columns: on the left, the evolution of PE according to TTOP; and on the right, according to ZAA. We hope the lines are now more clearly distinguishable. Regarding the reviewer's suggestion about enhancing compatibility between Figs. 8, 9, and 11, see R1C20.**

R1C26: REVIEWER'S COMMENT:

Figure 12 caption: Remove sentence: In this study ... 2.3.3). The caption should only describe the figure and its content.

**AUTHORS' RESPONSE:**

**The sentence has been removed from the figure's caption following the reviewer's indication.**

R1C27: REVIEWER'S COMMENT:

Figure 12: I suggest plotting the MPIESM-PePE members on top so that they can be better distinguished. Identifying single symbols is not necessary for the CMIP6 models so that they may be partially covered.

**AUTHORS' RESPONSE:**

**Thank you for the suggestion. In the submitted version of the manuscript, MPIESM-PePE members in Fig. 12 were already plotted on top of the CMIP6 markers. However, to further improve legibility following the reviewer's comment, we have increased the MPIESM-PePE and reduced the CMIP6 marker sizes, and deleted the black contours for CMIP6 markers. We hope this resolves the visibility issue.**

R1C28: REVIEWER'S COMMENT:

Line 524: ... profile *for 6*-8 % of

**AUTHORS' RESPONSE:**

**The text has been changed according to the reviewer's indication.**

R1C29: REVIEWER'S COMMENT:

Line 532-534: In fact, CMIP6 models *with* LSMs deeper than 40 m show the *smallest differences in ZAA-TTOP PE*, indicating that ESM simulations with deep LSM vertical schemes *are better at projecting* the future ...

**AUTHORS' RESPONSE:**

**The paragraph has been changed according to the reviewer's indications.**

R1C30: REVIEWER'S COMMENT:

Line 551: ... *Helmuth* Haak ...

**AUTHORS' RESPONSE:**

**The text has been changed according to the reviewer's indication.**

**2 Referee 2: Anonymous**

GENERAL COMMENTS:

R2C1: REVIEWER'S COMMENT:

The authors used a fully coupled MPI-ESM to study soil hydro-thermodynamics in permafrost regions. They examined the effects of soil hydro-thermal and depth on soil thermodynamics and permafrost distribution by comparing the default model setup (REF) and two extreme hydrological setups (WET and DRY), as well as the soil discretization setup (5 and 11 layers with approximately 10 m and 18 layers with 1391 m). The results suggest that the wide range of permafrost extent in CMIP6 is associated with soil depth and hydrological conditions. This well-designed paper falls within the scope of The Cryosphere and may be of interest to readers in the permafrost and modeling communities.

AUTHORS' RESPONSE:

The authors acknowledge the good perspective of the reviewer on the paper. Please find below the comprehensive point-to-point response to your review.

R2C2-a: REVIEWER'S COMMENT:

The manuscript is rather lengthy and could be written more concisely.

AUTHORS' RESPONSE:

We appreciate Reviewer 2's comment about the manuscript length and are sorry it gave the impression of being too long. We have revised the text carefully following all three reviewers' suggestions, focusing on making it clearer and more concise (R1C1, R2C2-c). We have also synthesized some discussions focused on quantitative results of different metrics with a table (see R1C3, R1C17, and R2C2-e). These changes have not drastically but slightly shortened the manuscript. However, we believe the sections are now more clearly explained and that the overall rationale comes across in a much more straightforward way. We hope the reviewer agrees with this. Otherwise, we are open to reconsider additional shortening, but at this stage we think the current draft represents a good compromise.

R2C2-b: REVIEWER'S COMMENT:

There are 12 figures that could be eliminated.

AUTHORS' RESPONSE:

We understand by this comment that the reviewer is concerned about the number of figures in the manuscript. We have carefully re-evaluated each figure and concluded that all of them are essential to present a complete and coherent picture of the study. This is also the opinion of Reviewer 1 (see R1C1). Figures 1–3 describe the model and experimental setups; Figures 4–6 assess permafrost temperature variability; Figures 7–9 show ALT

evolution; Figures 10 and 11 focus on PE; and Figure 12 explores the link between PE retreat and LSM depth. We therefore propose to keep all figures, but remain open to reconsidering this decision if the reviewer have specific concerns about individual figures.

R2C2-c: REVIEWER'S COMMENT:

Additionally, some sections, especially the introduction and methods, could be rearranged for clarity. For example, the simulation setups are introduced in the introduction, beginning at L85. This information could be moved to the methods section.

AUTHORS' RESPONSE:

We thank the reviewer for pointing this out. Indeed, the description of the more comprehensive soil hydrological features resolved by the modified version of JSBACH (JSBACH-HTCp) and the hydrological setups used in this work to test the sensitivity of permafrost to changes in soil wetness is well described in Section 2.1. Therefore, this information seems redundant and unnecessary in the introduction. Following the reviewer's suggestions, we have decided to remove the pieces of paragraph containing this redundant descriptions from Section 1, and keep them in Section 2.1.

R2C2-d: REVIEWER'S COMMENT:

The methods section contains several model abbreviations, such as MPI-ESM1.2, JSBACH3.2, JSBACH-HTCp, and MPIESM-PePE. This section should be simplified to avoid confusion.

AUTHORS' RESPONSE:

We have cautiously revised the amount and use of abbreviations in Section 2.1, and considered that all of them are necessary: MPI-ESM1.2-LR and JSBACH3.2 are the standard model and its land component, respectively; JSBACH-HTCp is the version of JSBACH3.2 with a more comprehensive representation of hydro-thermodynamic processes and coupling that we use in this work; and MPIESM-PePE is the ensemble of simulations with varying configurations of the hydrology and the vertical discretization presented in this work. Other abbreviations allude to other model components of the MPI-ESM1.2-LR and are well referenced. Nevertheless, to improve readability and considering the reviewer's concern, we removed minor version numbers from acronyms in the main text and figures (e.g., MPIOM instead of MPIOM6.3, JSBACH instead of JSBACH3.2). Additionally, to enhance readers' understanding (see R2C2-e), we reordered Section 2.1: first we describe the reference MPI-ESM and JSBACH configurations, then we detail the changes in JSBACH-HTCp, and finally we introduce the MPIESM-PePE ensemble. We have also ensured consistency in the use of abbreviations.

R2C2-e: REVIEWER'S COMMENT:

If the study used MPIESM-PePE with the JSBACH-HTCp land component, then MPI-ESM1.2 and JSBACH3.2 could be introduced first, and followed by an explanation of the improved features of JSBACH-HTCp and

MPIESM-PePE together with the experimental setup. Alternatively, add a table with the names and brief descriptions of the versions used for the simulations.

**AUTHORS' RESPONSE:**

**To avoid confusion and enhance readers' understanding, we have reordered the content in Section 2.1, following the reviewer's suggestion. In the revised version of the manuscript, we first present the reference MPI-ESM and JSBACH configurations, and then describe the changes made to JSBACH in JSBACH-HTCp, the LSM used to produce the MPIESM-PePE (see R2C2-d). We believe that the rationale is more clear now, so we consider that a table with the names and descriptions is not longer necessary. However, we are open to include it if asked by the reviewer.**

R2C2-f: REVIEWER'S COMMENT:

**Furthermore, the results and discussion sections sometimes contain a lot of values that could be summarized in a table, for example, the section 3.2.**

**AUTHORS' RESPONSE:**

**The large amount of numbers included in Section 3.2 concerning the evolution of ALT has been eliminated from the main text, and gathered into a new table (see Table 2 in the track changes or in the revised document), following the reviewer's indication. See also R1C3 and R1C17.**

SPECIFIC COMMENTS:

R2C3: REVIEWER'S COMMENT:

**L150: "Standard physics" sounds vague.**

**AUTHORS' RESPONSE:**

**Following the indication of the reviewer, we modified the text as follows (lines 169–170 of the track changes document):**

"... a third subset of experiments was run using the  physics of the original version of JSBACH ..."

R2C4: REVIEWER'S COMMENT:

**L258, L300: If the snow schemes differ between the WET/DRY (JSBACH-HTCp) and REF (JSBACH3.2) simulations, this should be mentioned in the methods section. Currently, it is unclear what differs between the WET/DRY and REF simulations. It should clearly describe how they differ, not only in the hydrological scheme (extreme wet and dry), but also in other schemes, such as the presence of an organic layer and water phase changes.**

**AUTHORS' RESPONSE:**

**WET and DRY configurations represent two hydrological extreme states attained setting different parameters within JSBACH-HTCp, whilst REF represents the reference state of the MPI-ESM LSM, JSBACH. A description of the differences between JSBACH and JSBACH-HTCp is given in Section 2.1, which is part of the methods section. However, following the reviewer's concern (see also R3C15), we have included a statement that explicitly clarifies the differences between the single-layer (JSBACH) and the multi-layer (JSBACH-HTCp) snow schemes (lines 157–159 of the track changes document):**

"A more realistic feature in JSBACH-HTCp in comparison with JSBACH is that both the soil water phase and thermal properties vary with temperature and moisture content, respectively. This allows for the existence of ice in Arctic soils, which is not represented by JSBACH. Other new hydrological features included with the JSBACH-HTCp comprise adapted interactions between soil hydrology and vegetation, with the inclusion of an organic topsoil layer, the possibility for supercooled water, an improved representation of percolation and drainage by including the effect of soil ice impedance, and the implementation of a simple wetland and a new multi-layer snow  scheme. This multi-layer scheme explicitly resolves thermal conductivity within the snow cover. By contrast, JSBACH uses a single-layer scheme that does not resolve internal snow temperature gradients, but only affects the computation of surface albedo and the thermal properties of the top soil layers. A thorough description of all changes incorporated in JSBACH-HTCp can be found in de Vrese et al. (2023)."

**Furthermore, we understand that due to the initial order in the submitted version of the manuscript, the description of the differences between JSBACH and JSBACH-HTCp was difficult to find and follow. We hope that it is more clear now as a result of the new structure of Section 2.1 (see R2C2-e).**

R2C5: REVIEWER'S COMMENT:

**L281: Increased soil depths (5 and 11 layers versus 18 layers) reduce the warming trend. However, REF18 appears warmer than REF5 and REF11, which may be due to their greater responsiveness to a cooler scenario (SSP1) compared to REF18. It would be better to describe that the deeper layers (18 layers) are less responsive to temperature change.**

**AUTHORS' RESPONSE:**

**The reviewer is right, REF18 shows greater warming trends for MAAT, MAGST, and MAST 5 m under the SSP1-1.9 scenario. However, note that the WET and DRY configurations do not show any substantial differences between 5L, 11L, and 18L discretizations for the same scenario, so REF differences are not conclusive. Moreover, small differences between discretizations, of similar magnitude to the one observed by the reviewer, are shown e.g., for the WET MAAT and MAGST under the SSP1-1.9 scenario, or the DRY MAGST and MAST 5 m under the SSP2-4.5. Such differences can be due to other reasons, like internal variability of the selected time intervals to calculate trends, but do not show a consistent behaviour that cal allow us to ascribe them to enhanced LSM**

depth. A clear example of consistent response is the smaller trend of MAST 5 m for all configurations (REF, WET, DRY) under the SSP2-4.5 and SSP5-8.5 scenarios, which is a sign of the amplitude attenuation of the trends as a response deepening the LSM. This was also observed for the same model in standalone simulations González-Rouco et al. (2021). To make this clear and in response to the reviewer's concern, we decided to modify the original sentence to (lines 331–337 of the track changes document):

"Regarding the effects of having different vertical discretizations, differences between the 5L, 11L, and 18L schemes can be identified across several configurations and scenarios. However, the only signal that is robust and consistent across REF, WET, and DRY is the attenuation of the MAST 5 m warming trend under the higher radiative forcing scenarios (SSP2-4.5 and SSP5-8.5).Deepening JSBACH systematically reduces the SSP5-8.5 warming trend at 5 m depth by approximately 0.1 °C dec$^{-1}$ at 5 m depth, indicating that a deeper LSM dampens subsurface temperature responses to surface warming."

R2C6: REVIEWER'S COMMENT:

L290: For the SSP5-8.5 scenario? Figures 4d and 4f show only temporal changes.

AUTHORS' RESPONSE:

Thank you for pointing this out. As indicated in the caption of Fig. 4, the temporal changes for the periods 2041–2060 and 2081–2100 shown in panels b, d, and f correspond to the SSP5-8.5 scenario. Note that the width of the boxplots represents the range of the spatial distribution of gridpoint values within the permafrost domain for the averages of a given period (1850–1900, 1995–2014, 2041–2060, or 2081–2100). This is what we called "permafrost domain boxplots" (Section 2.3.1). To make this more clear, we have modified the caption to:

"Temperature variability in the nine simulations of the MPIESM-PePE (see legend for colors) within the permafrost domain (Fig. 2a). (a) Time evolution (11-year moving average filtered outputs) of the median MAAT (dashed), MAGST (solid), and MAST 5 m (dotted line, ºC) in the PIC, HIS, and different SSP scenarios (see Section 2.2). Spatial distribution of MAAT (b), MAGST (d), and MAST 5 m (f) given by permafrost domain boxplots of time averages for the selected periods highlighted in (a) with light yellow bands: early-historical (1850–1900), late historical (1995–2014), mid-21st century (2041–2060), and late 21st century (2081–2100) in the SSP5-8.5 scenario. The inner tick in the boxplot marks the median value within the permafrost mask area (Fig. 2a), the bottom and top lines the quartiles 1 and 3, and the bottom and top whiskers the 10th and 90th percentiles, respectively. MAAT (c), MAGST (e), and MAST 5 m (g) boxplots indicating the range of trends (ºC dec$^{-1}$) within the permafrost domain, for the historical (HIS, 1850–2014) and SSP1-1.9, SSP2-4.5, and SSP5-8.5 scenario simulations (2015-2100) are also portrayed."

**R2C7: REVIEWER'S COMMENT:**

L347 (Fig. 7): What does "18L value is out of bounds" mean? Are there no data beyond this range? For the 18L simulations, can the maximum ALT be 9.98 m even though the soil layers exist deeper than that?

**AUTHORS' RESPONSE:**

**This sentence meant that the mid-layer depth of the bottom layer for the 18L discretization is out of bound of panels a–e (0–9 m), since it is of 1027.76 m. To make the sentence more clear, it has been modified to:**

"The mid-layer depth for the bottom layer of the 18L configuration lies outside the range shown in panels a–e and is therefore not included."

**Regarding the reviewer's question, in the 18L simulations, the maximum ALT can indeed exceed 9.98 m because the vertical profile extends deeper than that (layers 12 to 18). However, to facilitate a consistent comparison with the 5L and 11L vertical schemes, we mask any grid cells within the permafrost domain with ALT values greater than 7.86 m to this exact value. This depth corresponds to the mid-layer depth of the bottom layer in the 11L scheme and represents the maximum plausible ALT for that discretization. An explanatory note has been added to Section 2.3.2 (lines 230–233 of the track changes document):**

"For all vertical discretizations, the maximum physically resolvable ALT corresponds to the mid-layer depth of its bottom layer. To ensure comparability between schemes, ALT was capped to 7.86 m; any value above this threshold was set to 7.86 m. This limit matches the mid-layer depth of the bottom layer in the 11L configuration and thus represents the deepest meaningful ALT for that setup."

**R2C8: REVIEWER'S COMMENT:**

L396: What could cause these spatial variations?

**AUTHORS' RESPONSE:**

**The areas showing the earliest and most pronounced permafrost retreat (southern Siberia, southern Canada, the Labrador Peninsula, and Scandinavia) are located near the southern margin of the permafrost region, where surface conditions are generally warmer than in more northerly or higher-elevation areas of Siberia, Canada, and Alaska. These spatial patterns are therefore likely due to the fact that the southern permafrost edge experiences progressively longer periods of sustained temperatures near or above freezing with global warming, leading to widespread permafrost retreat and active layer thickening. Led by the reviewer's question, we have decided to include the following sentence in the manuscript (lines 459–461 of the track changes document):**

"These regions lie near the southern margin of the permafrost zone, where comparatively warmer surface conditions make them more sensitive to permafrost retreat and active layer thickening under climate change (e.g., Biskaborn et al., 2019; Rantanen et al., 2022)."

R2C9:   REVIEWER'S COMMENT:

 L404 (Fig. 10): The REF and WET simulations show a similar range, while the DRY simulations show slightly less permafrost extent, closer to the ESApCCIv3 value. Does this mean that the DRY simulations better represent the real world? If the REF and WET simulations overestimate the extent of permafrost, does that imply that current models (e.g., the REF simulation setup) are wetter than the real world?

AUTHORS' RESPONSE:

The WET and DRY configurations in this study are not designed to represent a more realistic state of Arctic permafrost than REF. They are idealized hydrological-end members, intended to sample uncertainty and to help explain part of the inter-model spread seen in state-of-the-art ESMs (see Section 2.1 and R3C3). It is also important to note that the ESApCCI product does not necessarily represent the true permafrost state. It is an observation-constrained reanalysis product, derived with the CryoGrid model driven by satellite land surface temperatures (see Section 2.3.3). The resulting permafrost extent is therefore conditional on CryoGrid's parameterizations and assumptions.

With this in mind, the fact that the DRY simulations happen to be closer to the ESApCCI estimate than REF or WET can have several explanations. For example, CryoGrid's representation of runoff, infiltration, and soil-ice impedance to drainage, hydrological processes controlling the permafrost water table state, could in combination yield a drier Arctic state that aligns more closely with DRY. However, these factors are difficult to assess, since the precise parameterizations used in CryoGrid for ESApCCI are not publicly available. Large-scale soil moisture data are also not given by the ESApCCI product. In conclusion, it is not possible to state that DRY is more realistic than REF or WET, nor that the Arctic permafrost system is systematically drier. Different processes or limitations in the realism of the simulations can balance out and make them agree with observations.

We consider that ESApCCI dataset is valuable as a reference to depict how far our simulations can be from observation-constrained model estimates, but agreement with it should not be over-interpreted as a direct measure of realism. A paragraph acknowledging this circumstance has been included in Section 2.3.3 (lines 266–267, 277–282 of the track changes document):

"In order to illustrate the magnitude of the differences between the variety of configurations and vertical discretizations used in this work, theThe results for the MPIESM-PePE are compared with the ZAA and TTOP PE estimates stemming from Steinert et al. (2023) for an ensemble of 34 CMIP6 HIS and SSP5-8.5 experiments. Also, data provided by the Permafrost Climate Research Data Package version 3 product of the European Space Agency Climate Change Initiative database (ESApCCIv3) are used as an illustrative reference. ESApCCIv3 delivers mid- to high-latitude Northern Hemisphere (north of 30º) gridded data of permafrost [...] is used to compute the corresponding PE. It is important to remark that, while ESApCCIv3 provides a valuable pan-Arctic benchmark of permafrost state, it remains a model product with known biases in the representation of permafrost temperature, ALT, and PE (Obu et al., 2021). For this reason, it is used as a reference to illustrate the order of magnitude of PE differences between the different hydrological configurations

and vertical discretizations presented in this study. However, this comparison does not attempt to provide an assessment of whether any of the MPIESM-PePE simulations are better or closer to the real Arctic permafrost state."

**Also, see some discussion about the comparison with ESApCCIv3 in Section 3.3 (lines 492–498 of the track changes document):**

"... In fact, state-of-the-art ESMs have been claimed to have difficulties in reproducing nowadays PE, due to their limitations in representing land surface processes (Chadburn et al., 2015; Burke et al., 2020; Steinert et al., 2023), the non-inclusion of specific processes accelerating permafrost degradation at local scales, such as soil subsidence (Andresen et al., 2020), talik formation or thermokarst (Farquharson et al., 2019; Nitzbon et al., 2024), or the inability to account for the permafrost-carbon warming feedback (Burke et al., 2013). These observation-simulation discrepancies in PE are suggested in Fig. 10, which exhibits that PE values from ESApCCIv3 in 1997–2019 are not only lower than the CMIP6 median, but also lower than all the MPIESM-PePE experiments, both for ZAA and TTOP. The differences are especially large for ZAA, which might be more sensitive to the aforementioned degradation processes enhancing deep permafrost losses that are not implemented in current LSMs, including JSBACH. However, the observation–simulation comparison presented here must be interpreted with caution. The ESApCCIv3 product is not a purely observational dataset but an assimilation product, and it has documented biases in the representation of permafrost temperature, ALT, and PE (Obu et al., 2021). Nonetheless, the broad spread in PE values across ESApCCIv3, CMIP6 models, and MPIESM-PePE experiments illustrates the substantial uncertainty that still surrounds our knowledge of the current permafrost state, of which soil hydrology is only one contributing factor."

**and in the conclusions (lines 613–618 of the track changes document):**

"Finally, the comparison of PE values from the MPIESM-PePE with observation-constrained ESApCCIv3 estimates and other CMIP6 simulations illustrates how the range of permafrost response that can stem in one model from different assumptions in permafrost hydrology contributes to model uncertainties. Furthermore, observation-constrained products based on data assimilation, i.e., models constrained by observational evidence, are not free from model biases resulting from the decisions on parametrizations or physical processes included. The latter emphasizes the need fore more observations and assessment of uncertainties in observation-based products."

R2C10: REVIEWER'S COMMENT:

L455: Again, what could be causing these spatial variations?

**AUTHORS' RESPONSE:**

**The main reason is the warmer surface conditions in these areas (see R2C8). We included a clarifying comment to stress this statement (lines 526–528 of the track changes document):**

"Permafrost degradation is more intense in southern Siberia, Fennoscandia, and the Labrador Peninsula, warmer areas lying near the southern margin of the permafrost zone. By 2041–2060,  only a quarter of the CMIP6 members and REF18, WET11, and WET18 still show some permafrost patches in southern Siberia  (Fig. 11c,g)".

**3  Referee 3: Anonymous**

GENERAL COMMENTS:

R3C1:  REVIEWER'S COMMENT:

In their manuscript, García-Pereira et al. perform historical and future simulations with the MPIESM, which was modified to account for a dry and wet configuration of Arctic soils, as well include various model soil depths. With the help of these simulations, the authors investigate the differences of these setups for soil surface temperature, active layer depths and permafrost extent for past and future scenarios. The work is an important effort and relevant for the journal Cryosphere. I believe the work is close to publication, but I have a few major comments that, in my opinion, should be addressed.

AUTHORS' RESPONSE:

The authors acknowledge the good perspective of the reviewer on the paper. Please find below the comprehensive point-to-point response to your review.

MAJOR REMARKS:

R3C2:  REVIEWER'S COMMENT:

I believe the manuscript can be somewhat stream-lined to stress novel results from this specific study. Some of the extensively discussed points are already addressed in other papers (e.g. aspects of the importance of freeze-thaw processes and soil hydrology for the Arctic heat and water balance). There is also some very technical information on model versions and reasoning for these changes since MPI-ESM of CMIP6 simulations (which is not applied here), which I believe is maybe not that interesting to readers not directly involved with the model (and would be anyways better fitted for a model development paper). I would probably just focus on what is present in the model versions presented here.

AUTHORS' RESPONSE:

We thank the reviewer for this suggestion. We agree that the manuscript benefits from being more concise and from placing greater emphasis on the specific novelties of this study. Following the reviewers' comments, we have revised the text to streamline the abstract (see changes in R1C2) and discussion (see changes in R2C2-a, R2C4, R2C5, R2C8), reduce background that is already well covered in the literature (see changes in R2C2-c), and describe more concisely the model and the experimental setup used in this work (see changes in R2C2-d, R2C2-e). We have also tried to convey more clearly the conclusions that arise from the analysis of the experiments and

its significance (e.g., R3C7). We hope that in its present form the manuscript highlights more directly the main findings and the novel aspects of our work.

R3C3: REVIEWER'S COMMENT:

Inclusion of observational constraints: The authors compare modelled permafrost extent to an observational-derived product (with moderate/poor agreement for both dry and wet configurations), but do not compare soil temperatures and active layer depths shown here with similar available products. I recommend consistently including ground temperature and active layer thickness observational products as well (https://climate.esa.int/en/projects/permafrost/), to better grasp the model performance in simulating soil temperature and active layer depths. Since these datasets are also based on thermal models and limitations should probably also be briefly mentioned.

AUTHORS' RESPONSE:

Thank you for this suggestion. We are aware that ground temperature and active layer thickness exist for ESApCCI, and we agree they are useful. We reflect this in the new version of the manuscript, also noting that they are model-based and no free from limitations. However, in this study we included only the permafrost extent product as a single reference to illustrate the order of magnitude of the differences in the mean state between configurations (WET/DRY) and vertical discretizations (5L/11L/18L). Our goal was not to provide an evaluation of model results but to highlight the scale of the changes in comparison with realistic observation-constrained estimates. Acknowledging the reviewers' concern, we have included a remark in that sense (see also R2C9). If the inclusion of this observational product is misleading or invites captious interpretation, we are willing to remove it or move it to the Supplement to avoid over-stating model performance.

R3C4: REVIEWER'S COMMENT:

The simulation setups presented are idealized to potentially represent dry and wet soils, and it is unclear which set up is actually more realistic to represent the Arctic as a whole, both from an observational standpoint as well as in terms of process representation. Since there are both relatively dry and wet areas in the Arctic, would it be possible to combine the results from the wet and dry setup offline, using wetland coverage maps to weight every grid cell to give an estimated combined of permafrost thaw in the future?

AUTHORS' RESPONSE:

The WET and DRY configurations in this study are not meant to represent a more realistic state of Arctic permafrost than REF. They are idealized end-members of permafrost hydrology intended to sample uncertainty and help explain part of the inter-model spread shown by state-of-the-art ESMs (see Section 2.1 and R2C9). From this perspective, although some Arctic regions will likely become drier with warming while others become wetter, our experiments do not aim to capture this spatial variability or to represent a best match with observations.

Following this line of reasoning, it would be in principle technically feasible to combine the two configurations offline, for example, using a "chessboard" approach that applies the WET or DRY parametrization by grid cell based on a wetland coverage map. However, this is outside the scope of the present work and we would need to consider the feasibility of its implementation. Nevertheless, we acknowledge the idea and agree it is an interesting possibility.

R3C5: REVIEWER'S COMMENT:

A bit of an open question: the authors stress the differences in permafrost extents by the different approaches/simulation setups, which I think is fair to grasp a magnitude of impact that can be communicated as a single number. However, the differences in permafrost extent likely are differences of areas of very deep permafrost (I assume mostly >10m depth?), which would be probably disconnected from surface fluxes and at which depths I would assume that mostly bedrock or mineral soil is found. Is the thawing in these depths really relevant for a discussion on climate impacts of permafrost on surface hydrological feedbacks for the climate and carbon degradation? In my opinion the importance deepening of close-to-surface carbon-rich layers (such as the Yedoma domain) should be stressed, even if this is not reflected in a full permafrost retreat of the soil column.

AUTHORS' RESPONSE:

Thank you for this thoughtful comment. Although one of the permafrost definitions we used (ZAA, i.e., permafrost presence at the depth of zero annual amplitude) is often described as a "deep" permafrost diagnostic, in our context it does not target changes at great depths that respond only on multi-centennial to millennial timescales, well beyond the duration of our simulations. In fact, two of our three vertical discretizations (5L and 11L) have an LSM depth of around 10 m (9.83 m and 9.98 m) and therefore cannot resolve deep permafrost. Our aim when we compare PE between the different discretizations is to illustrate how PE differences can arise from imposing too-shallow lower boundary conditions (zero flux at 10 m). Under sustained surface warming, too shallow LSM boundary conditions introduce a warm bias in the deepest modeled layers. Deepening the column (18L) removes this artificial bottom boundary influence and yields a more physically consistent permafrost temperature, and subsequently ALT and PE; evolution. While the bias is smaller near the surface, it can still propagate upward and affect soil hydrology and the deepest modeled carbon pools (within the upper 10 m), which motivates the deeper column for coupled projections.

In short, our message is not that loss of very deep permafrost dominates climate impacts, but that using a deeper LSM improves the physical representation of permafrost variability (temperature, ALT, and PE) by avoiding boundary condition artifacts. Following the reviewer's suggestion and to enhance the relevance of LSM depth in deep permafrost, we have included a sentence acknowledging the impacts of LSM depth on deep soil organic carbon deposits (i.e., Yedoma, lines 610–612 of the track changes document):

"Regarding the LSM depth, the 5L and 11L with respect to the 18L discretization show consistent PE differences in the three hydrological configurations, of 1.9–3.1 $10^6$ km$^2$. This emphasizes the relevance of LSM depth in reducing deep permafrost degradation. In fact, CMIP6 models with LSMs deeper than 40 m show the smallest differences in ZAA – TTOP PE, indicating that ESM simulations with deep LSM vertical schemes are better at projecting the future degradation of deep permafrost and its sensitivity to surface warming. Furthermore, because ice-rich, carbon-bearing deposits (e.g., Yedoma) can extend well below 10 m, using deeper LSMs reduces bottom-boundary warming and leads to more realistic estimates of the vulnerability of these deep soil organic carbon reservoirs."

R3C6:  REVIEWER'S COMMENT:

It is not completely clear to me from the abstract and conclusions, what the final message was regarding the effects of the difference model soil depth configurations on the trends of averaged active layer thickness and PE. Looking at the effects of various model soil depths in Figure 4, 6 and Figure 10, this effect seems small (and probably not worth it in terms of additional computational expenses for coupled models?).

AUTHORS' RESPONSE:

We understand the reviewer's caveats concerning the role of LSM depth in the mean state and evolution of permafrost temperature, active layer thickness, and extent. Although the main impacts of deepening the LSM were stated in the abstract and conclusions of the submitted version, the writing there was perhaps not sufficiently clear and may have obscured the message. We have now thoroughly revised the manuscript (see R1C2, R1C3, R1C17, R2C2), including a clearer description of the effects of deepening the LSM on permafrost temperature trends (see R2C5) and a more explicit explanation of how ALT is characterized in the 5L, 11L, and 18L simulations (see R2C7). We also improved the rationale in the abstract and conclusions (see R1C2 and R3C7). We hope the effects of LSM depth are now clearer. For instance, in the abstract:

"Deepening the LSM reduces near-surface soil warming by about 0.1 °C dec$^{-1}$ in high radiative forcing scenarios, reducing permafrost retreat by up to 1.9–3.1 $10^6$ km$^2$ by the end of the 21st century."

Even though the effects of deepening the LSM are small, they might be of importance for the simulations of deep permafrost, rich in Yedoma (see R3C5). Moreover, the differences in terms of computation are not so large. Whilst a year of simulation of the standard fully-coupled MPI-ESM model (shallow 5L) costs 2.2 node hours on 4 nodes, the cost for a year of 18L simulation is just of 2.4 node hours on 4 nodes. Thus, for less than 10 % increase in computational cost, a much more realistic representation of permafrost thermodynamics is achieved, which we consider a reasonable trade-off for LSM-focused studies.

**R3C7: REVIEWER'S COMMENT:**

In terms of significance, would it be worth to include compute back-of-the-envelope calculations implications for carbon release from the permafrost (e.g. from typical spatial carbon contents?).

**AUTHORS' RESPONSE:**

**Thank you for this suggestion. We have considered it and believe that translating permafrost extent estimates (Section 3.3) into back-of-the-envelope estimates of the soil organic carbon that becomes vulnerable to release from permafrost thaw might be of much significance for the conclusions of the study. We find especially interesting yielding differences in vulnerable carbon between WET and DRY. Therefore, we have included a paragraph in the discussion about the differences of WET and DRY in terms of potential carbon release, following the reviewer's suggestion (lines 600–605 of the track changes document):**

"The implications of soil hydrology and LSM depth are also remarkable for the evolution of the PE. Both TTOP, which physically represents near-surface PE, and ZAA, which tracks deep PE evolution, yield an initial PE of around 15 $10^6$ km$^2$ for DRY and 20 $10^6$ km$^2$ for REF and WET, respectively. These differences explain up to 76 % of the interquartile range for CMIP6 deep PE, 54 % for near-surface PE, and about a quarter of the interdecile range for both, highlighting the uncertainty in current PE estimates associated with the representation of soil hydrology. Considering that permafrost region contains 1035 $\pm$ 150 PgC over 17.6 $10^6$ km$^2$, which corresponds to an areal mean of 58.8 $\pm$ 8.5 $10^{-6}$ km$^{-2}$ PgC (Hugelius et al., 2014), the differences between WET and DRY near-surface PE values would roughly amount for 250–340 PgC. This is around 40 % of the carbon currently stored in the atmosphere, a large fraction that indicates the importance of uncertainties in frozen-soil hydrology for permafrost carbon storage, with high-impact implications for the permafrost–carbon feedback and, in turn, climate change."

MINOR REMARKS:

**R3C8: REVIEWER'S COMMENT:**

L24: nearly four times?

**AUTHORS' RESPONSE:**

**The text has been changed according to the reviewer's suggestion.**

**R3C9: REVIEWER'S COMMENT:**

L48: I think PE is not defined anywhere.

**AUTHORS' RESPONSE:**

**The reviewer is right, so "PE" has been changed to "permafrost extent (PE)" to define this acronym, as it is the first time it appears in the manuscript.**

R3C10:  REVIEWER'S COMMENT:

 L70: yedoma -> Yedoma

**AUTHORS' RESPONSE:**

**The text has been changed according to the reviewer's indication.**

R3C11:  REVIEWER'S COMMENT:

 L74: Also the QUINCY model:

Lacroix, F., Zaehle, S., Caldararu, S., Schaller, J., Stimmler, P., Holl, D., Kutzbach, L., & Göckede, M. (2022). Mismatch of N release from the permafrost and vegetative uptake opens pathways of increasing nitrous oxide emissions in the high Arctic. Global Change Biology, 28, 5973–5990. https://doi.org/10.1111/gcb.16345

**AUTHORS' RESPONSE:**

**A mention to the QUINCY model (Lacroix et al., 2022) has been included, following the reviewer's indication (line 90 of the track changes document).**

R3C12:  REVIEWER'S COMMENT:

 L118: "An enhanced vertical resolution accounts for a better representation of hydrothermodynamic processes near the surface (Chadburn et al., 2015)." -> Reference to your Table 1?

**AUTHORS' RESPONSE:**

**A reference to Table 1 has been included after the cite to Chandburn et al. (2015), following the reviewer's suggestion.**

R3C13:  REVIEWER'S COMMENT:

 L188: "In case this depth does not specifically match a certain mid-layer depth value, JSBACH yields ALT using linear interpolation."

When would this be the case? I would imagine this would apply if a layer is partly frozen, but the authors explicitly define the >273.15 °C definition.

**AUTHORS' RESPONSE:**

**JSBACH discretizes the soil column into progressively deeper layers and solves the heat conduction equation using**

a Richtmyer–Morton numerical scheme (Reick et al., 2021). Temperatures are diagnosed at the mid-layer depths of each layer. When the depth of the 273.15 K isotherm (used to define ALT) does not coincide with a mid-layer depth, which is generally the case, its position is obtained by linear interpolation between the two adjacent layers. For example, if layer i is at 273 K and layer i+1 at 274 K, the 273.15 K depth is found by interpolating between their mid-layer depths in proportion to the temperature difference (i.e. adding to the depth of layer i a fraction of the distance to layer i+1 given by $(273.15 - 273)/(274 - 273)$). So, to make this more clear in the manuscript and following the question of the reviewer, the sentence:

"In case this depth does not specifically match a certain mid-layer depth value, JSBACH yields ALT using linear interpolation"

has been modified to (lines 228–230 of the track changes document):

"If the depth of the 273.15 K isotherm does not precisely coincide with a given mid-layer depth, JSBACH computes ALT by linearly interpolating between the temperatures of the two adjacent layers that are immediately warmer or cooler than this isotherm."

R3C14: REVIEWER'S COMMENT:

L293: Define winter offset in the text and why it is important.

AUTHORS' RESPONSE:

Winter offset is already defined in Section 2.3.1 as follows:

"In this paper, the winter (summer) offset is computed as the ground surface temperature at 0.1 m depth (GST) vs. surface air temperature (SAT) differences (GST-SAT) in December-January-February (June-July-August), following a similar approach to Burke et al. (2020)."

A sentence explaining why assessing winter and summer offset is important was included after the aforementioned sentence (lines 221–223 of the track changes document):

"Winter and summer offsets govern seasonal ground–air coupling, so evaluating them helps identify model biases in the representation of present-day and future projections of permafrost temperature and ALT."

Following the reviewer's indication, a reference to Section 2.3.1 where the definition of winter offset is given has been included (see line 347 of the track changes document).

R3C15: REVIEWER'S COMMENT:

L301: "Both results reflect weaker insulation in the standard snow model." Is this because of different parameterizations of the snow model (heat constants?), or added processes?

Wouldn't it be possible to back this up with data from the simulation?

**AUTHORS' RESPONSE:**

Thank you for the question. The weaker insulation in REF is due to the snow model. DRY and WET use a multi-layer snow scheme, which resolves vertical heat conduction (see R2C4). As thermal diffusivity of snow is low, heat conduction within the pack is hindered, resulting in a amplitude attenuation of the atmospheric temperature changes when transmitted to the soil. In contrast, REF uses a bucket scheme, which does not resolve internal snow temperature gradients and only affects albedo and the thermal properties of the top soil layers (see R2C4). As a result, atmospheric temperatures are poorly attenuated and winter and summer offsets are much smaller than in DRY and WET (see Figs. 5a and 6a). We back this up with simulation data of snow depth vs. winter offset (see Fig. 5) and explain the aforementioned differences in the next paragraph (lines 359–379 of the track changes document). Therefore, by the reviewer's comment, we understand that this sentence is ambiguous where it stands in the text and we decided to remove it from the paragraph.

R3C16:  REVIEWER'S COMMENT:

Figure 1. I know this is a conceptual figure, but the bedrock seems very deep? The bed rock in the Arctic is usually found around 1-2 m (unless in valleys).

**AUTHORS' RESPONSE:**

Yes, bedrock in most permafrost regions of the Northern Hemisphere is indeed very shallow, typically within the first 1–2 m of the soil column. In fact, the soil-bedrock mask imposed to JSBACH as a fixed field to define the depth range for soil hydrology does not reach a depth of 1 m in most gridcells. However, to keep the conceptual style consistent between Fig. 1 and 2b, and because Fig. 2b already contains many text boxes and arrows representing soil processes and parameters, we intentionally exaggerated the bedrock depth in Fig. 1. We have added a note to the captions of Fig. 1 and 2 to acknowledge this schematic exaggeration:

[revised manuscript text omitted]

---

## Author Response (AR2)

**Response to the reviewers' comments, egusphere-2025-2126, García-Pereira et al., revision round 2**

Félix García-Pereira1,2, Jesús Fidel González-Rouco1, Nagore Meabe-Yanguas1, Philipp de Vrese3, Norman Julius Steinert4, Johann Jungclaus3, and Stephan Lorenz3

The authors would like to thank the reviewers for their constructive suggestions and the time they devoted to reading and evaluating the manuscript. We have tried to integrate all suggestions and think that the manuscript has improved with them. We do appreciate their contribution.

The next sections contain a detailed point-by-point response to the reviewers' comments. Comments are labeled by reviewers and in order of appearance, i.e. R2C3 is the third comment of reviewer 2. The original number by the reviewer is also preserved if it was given.

**1 Referee 1**

**AUTHORS' COMMENT:**

The authors welcome the very positive perspective of Dr. Hagemann on the paper in this second round of reviews, and are very grateful for his comments during the first phase of the review process.

**2 Referee 3: Anonymous**

**GENERAL COMMENTS:**

**R3C1: REVIEWER'S COMMENT:**

I believe the authors provide very convincing responses to the reviews and improved the manuscript. I would still appreciate one or two sentences on the model version performance and biases, even from past studies, since it would might still help situate the thawing response of the model configurations, even though I understand the goal is to represent the effects of hydrological and LSM depth variability between models in an idealized manner (although it is an open question if dry/wet differences would be the same from model to model).

<sup>1Geosciences Institute, IGEO (CSIC-UCM), Madrid, Spain

<sup>2Faculty of Physical Sciences, Complutense University of Madrid (UCM), Madrid, Spain

<sup>3Max Planck Institute for Meteorology, Hamburg, Germany

<sup>4CICERO - Center for International Climate Research, Oslo, Norway

**AUTHORS' RESPONSE:**

The authors acknowledge the good perspective of the reviewer on the paper in this second round of reviews. Concerning the request for a brief statement on model performance and biases the authors agree that, while the main limitation of JSBACH, i.e., the omission of hydro-thermodynamical coupling (no soil moisture phase changes and static soil thermal properties), is discussed in Section 2.1, the consequences in terms of performance and biases are not explicitly mentioned. We have tried to address this issue including the following text (lines 96–99 of the track changes document):

"The standard version of JSBACH (Fig. 1) does not include any coupling processes between soil thermodynamics and hydrology, which prevents soil ice from forming under freezing conditions. This limitation leads to overly warm soils and thawing depths in summer, contributing to a warm bias over high latitude continental areas. This limitation issue was addressed by Ekici et al. (2014) and Steinert et al. (2021b) by ..."

For the additional minor remarks, please find below the comprehensive point-to-point response.

**MINOR REMARKS:**

**R3C2: REVIEWER'S COMMENT:**

L58 Maybe a fitting additional reference here is Peng et al., 2023: https://doi.org/10.1029/2023EF003573

**AUTHORS' RESPONSE:**

A reference to the Peng et al. (2023) has been included, following the reviewer's indication (line 58 of the track changes document):

"Moreover, as shown by Chadburn et al., 2015, a more realistic representation of these processes within LSMs can lead to estimates of ALT and PE that are closer to the latest observation-constrained estimates (Chadburn et al., 2015; Peng et al., 2023)."

**R3C3: REVIEWER'S COMMENT:**

L127 Further, a more realistic deeper LSM depth permits an unbiased representation of subsurface temperature variability and land heat storage. -> "unbiased" -> improved?

**AUTHORS' RESPONSE:**

The text has been changed according to the reviewer's indication.

**R3C4: REVIEWER'S COMMENT:**

L172 "Winter and summer offsets govern seasonal ground-air coupling, so evaluating them helps identify model

biases in the representation of present-day and future projections of permafrost temperature and ALT." But this evaluation is not actually done in this study? I would then remove this.

**AUTHORS' RESPONSE:**

Although winter and summer offsets are indeed evaluated in Figs. 5 and 6 and discussed in Section 3.1, we agree that such a strong statement as "Winter and summer offsets govern seasonal ground-air coupling" would need a thorough evaluation of the spatio-temporal seasonal response of surface air vs. ground surface temperatures at different seasons, which is out of the scope of this paper. Recent literature (e.g., Nitzbon et al., 2025, in review) evidences the impacts of seasonal temperature amplitude on maximum thaw depths and permafrost cracking at paleoclimate scales. For this reason, the authors have decided to degrade the strength of the first sentence, and also add some supportive references:

"Winter and summer offsets governinfluence seasonal ground–air coupling (Melo-Aguilar et al., 2018; de Vrese et al., 2023), so evaluating them helps identify model biases in the representation of present-day and future projections of permafrost temperature and ALT (Nitzbon et al., 2025)."

**R3C5: REVIEWER'S COMMENT:**

L231 "For this reason, it is used as a reference to illustrate the order of magnitude of PE differences between the different hydrological configurations and vertical discretizations presented in this study. However, this comparison does not attempt to provide an assessment of whether any of the MPIESM-PePE simulations are better or closer to the real Arctic permafrost state." These explanations, especially the first part are not really clear. If it is important, maybe try to break it down.

**AUTHORS' RESPONSE:**

We thank the reviewer for pointing this out. Based on reviewers' comments R2C9 and R3C3 in the first round of reviews, we considered it important to explicitly state that ESApCCIv3 is used only as a reference to place the magnitude of PE differences between the different MPIESM-PePE configurations and vertical discretizations in context. For this reason and following the reviewer's suggestion, we have decided to keep this paragraph but have revised it for clarity as follows:

"For this reason, itthis product is not used as ground truth of the current PE, but as a reference to illustrate the order of magnitude of PE differences between the different hydrological configurations and vertical discretizations presented model versions used in this study. However Therefore, this the MPIESM-PePE vs. ESApCCIv3 PE comparison does not attempt to provide an assessment of determine whether any of the MPIESM-PePE simulations are better or closer to the real Arctic permafrost state."

**R3C6: REVIEWER'S COMMENT:**

L517 "These differences are mainly due to their different representation of soil thermodynamics and hydrology

(Andresen et al., 2020)." I feel "mainly" might be too strong here, would model differences in Arctic amplification not have a stronger effect to the spread of air temperature? Models may not represent the high rate of warming in the Arctic correctly (Rantanen et al., 2022).

**AUTHORS' RESPONSE:**

We believe that the reviewer is definitely right with this point, and we thank him for this appreciation. In order to weaken our statement, we have replaced "are mainly due to" by "are strongly influenced by", to comply with the reviewer's suggestion.

**R3C7: REVIEWER'S COMMENT:**

L15 "can be partly due" -> can be partly explained

**AUTHORS' RESPONSE:**

The text has been changed according to the reviewer's indication.

**R3C8: REVIEWER'S COMMENT:**

L15 this -> our findings

**AUTHORS' RESPONSE:**

The text has been changed according to the reviewer's indication.

**References**

- Chadburn, S. E., Burke, E. J., Essery, R. L. H., Boike, J., Langer, M., Heikenfeld, M., Cox, P. M., and Friedlingstein, P.: Impact of model developments on present and future simulations of permafrost in a global land-surface model, The Cryosphere, 9, 1505–1521, https://doi.org/10.5194/tc-9-1505-2015, 2015.
- de Vrese, P., Georgievski, G., González Rouco, J. F., Notz, D., Stacke, T., Steinert, N. J., Wilkenskjeld, S., and Brovkin, V.: Representation of soil hydrology in permafrost regions may explain large part of inter-model spread in simulated Arctic and subarctic climate, The Cryosphere, 17, 2095–2118, https://doi.org/10.5194/tc-17-2095-2023, 2023.
- Ekici, A., Beer, C., Hagemann, S., Boike, J., Langer, M., and Hauck, C.: Simulating high-latitude permafrost regions by the JSBACH terrestrial ecosystem model, Geoscientific Model Development, 7, 631–647, https://doi.org/10.5194/gmd-7-631-2014, 2014.
- Melo-Aguilar, C., González-Rouco, J. F., García-Bustamante, E., Navarro-Montesinos, J., and Steinert, N.: Influence of radiative forcing factors on ground–air temperature coupling during the last millennium: implications for borehole climatology, Climate of the Past, 14, 1583–1606, https://doi.org/10.5194/cp-14-1583-2018, 2018.
- Nitzbon, J., Langer, M., Müller-Ißberner, L. A., Dietze, E., and Werner, M.: How temperature seasonality drives interglacial permafrost dynamics: Implications for paleo reconstructions and future thaw trajectories, EGUsphere, 2025, 1–35, https://doi.org/10.5194/egusphere-2024-4011, 2025.
- Peng, X., Zhang, T., Frauenfeld, O. W., Mu, C., Wang, K., Wu, X., Guo, D., Luo, J., Hjort, J., Aalto, J., Karjalainen, O., and Luoto, M.: Active Layer Thickness and Permafrost Area Projections for the 21st Century, Earth's Future, 11, e2023EF003573, https://doi.org/10.1029/2023EF003573, 2023.
- Steinert, N. J., González-Rouco, J. F., de Vrese, P., García-Bustamante, E., Hagemann, S., Melo-Aguilar, C., Jungclaus, J. H., and Lorenz, S. J.: Increasing the depth of a Land Surface Model. Part II: Sensitivity to improved coupling between soil hydrology and thermodynamics and associated permafrost response, Journal of Hydrometeorology, 22, 3231 3254, https://doi.org/10.1175/JHM-D-21-0023.1, 2021b.